# An Arctic sea ice concentration data record on a 6.25 km polar stereographic grid from three-years' Landsat-8 imagery

Hee-Sung Jung[1], Sang-Moo Lee[1,2], Joo-Hong Kim[3], and Kyungsoo Lee[1]

[1]Department of Earth and Environmental Sciences, Seoul National University, Seoul, 08826, Republic of Korea
[2]Institute for Data Innovation in Science, Seoul National University, Seoul, 08826, Republic of Korea
[3]Korea Polar Research Institute, Incheon, 21990, Republic of Korea

*Correspondence to*: Sang-Moo Lee (sangmoolee@snu.ac.kr)

**Abstract.** The decline of Arctic sea ice in the global warming era has received much attention as a contributing factor to the changes in the weather/climate in the Arctic and beyond. The coverage of Arctic sea ice (i.e., sea ice concentration (SIC))
has been monitored since 1972 using satellite passive microwave (PMW) measurements because of their extensive coverage and all-weather capability. However, the fundamental basis of algorithms for estimating SIC has not improved much since the early days due to the lack of reference SIC data, leading to discrepancies between existing PMW SIC algorithms. To overcome this issue, this study aims to construct data records of reference SIC over Arctic sea ice using 30 m resolution imagery from the Operational Land Imager (OLI) onboard Landsat-8. In order to collect relatively bright and clear scenes,
thresholds of solar elevation > 15° and cloud cover < 10% were applied in this study. Clouds in each Landsat-8 scene were masked using the cloud masking array provided in Landsat data. Due to the poor accuracy of the cloud masking array over ice-covered surface types, an additional step of visually inspecting the state of cloud mask using the true-colour image was designated in this study. Each Landsat-8 scene was sorted into four categories depending on the state of cloud mask. Normalized Difference Snow Index and OLI band 5 reflectivity were used to differentiate between ice and open water within
each selected Landsat-8 pixel. The classified data were projected onto a 6.25 km polar stereographic grid, and SIC for each grid cell was obtained by counting ice-classified pixels within the grid. SIC was only computed for grid cells with more than 99% of its area covered with Landsat-8 pixels to limit the uncertainty in SIC arising from grids that are not fully concentrated with Landsat-8 pixels. Uncertainty in the produced SIC was 1~4%, inferred using the Gaussian error propagation method. Out of 15,286 collected Landsat-8 images, 14,297 images were translated into SIC maps, and a total of
2,934,399 Landsat-8 SIC grid cells were generated. Evaluation of Landsat-8 SIC with SIC from ice charts revealed a good linear relationship (correlation coefficient of 0.96) between the two products as well as a mean negative bias which fell within the uncertainty range of Landsat-8 SIC. SIC based on Landsat-8 can be used as reference SIC to evaluate existing SIC products and thus one can improve SIC products as well as the use of the improved SIC for other applications such as data assimilation and retrieval studies. The vast amount of Landsat-8 SIC generated in this study may also be used to train deep
learning models for estimation of Arctic SIC coverage. The Landsat-8 SIC dataset can be publicly accessed at

## 1 Introduction

Since space-borne multi-channel passive microwave (PMW) observations have been available, areal information of Arctic sea ice has been successfully monitored. During the past four decades, these observations have shown that sea ice extent (SIE) which is defined as the area of ocean where sea ice concentration (SIC) is greater than 15% has been rapidly declining at a statistically significant negative trend of -12.7% per decade observed in September (Cavalieri and Parkinson, 2012; Meier et al., 2014; Meier and Stroeve, 2022). In the global warming era, the change in Arctic sea ice area is considered as a key indicator of climate change and this is closely associated with the changes in the Arctic local weather as well as the weather at mid-latitudes (Honda et al., 2009; Jaiser et al., 2012; Kim et al., 2014; Trewin et al., 2021; Shi et al., 2023). Therefore, obtaining precise observations of Arctic SIC is essential in order to diagnose the influences of climate change on Arctic sea ice.

As mentioned above, the spatial coverage of Arctic sea ice (i.e., SIC) has been monitored using satellite PMW measurements with their extensive spatial coverage over the Arctic and all-weather capability. Beginning with the launch of the Electrically Scanning Microwave Radiometer (ESMR) onboard Nimbus-5 in 1972 (Parkinson et al., 1987), successive launches of PMW sensors has allowed for the construction of comprehensive and continuous records of Arctic SIC. The Scanning Multi channel Microwave Radiometer (SMMR), launched in 1978, was equipped with five channels (6.6, 10.7, 18.0, 21.0, and 37.0 GHz) in first two Stokes' polarizations. The emergence of multi-channel PMW radiometers has led to the development of various SIC retrieval methods which were more accurate relative to the previous methods used for ESMR which only had a single channel at 19 GHz. Addition of the near 90 GHz high frequency channels in the PMW sensors following SMMR, which include the Special Sensor Microwave Imager (SSM/I), the Special Sensor Microwave Imager/Sounder (SSMIS), the Advanced Microwave Scanning Radiometer - Earth Observing System (AMSR-E), and the AMSR2, has allowed for spatially enhanced SIC retrievals.

Various PMW SIC algorithms have been developed, which estimate SIC based on combinations of brightness temperatures (TB) at various channels and empirically derived tie-points. One of the best-known algorithms is the Bootstrap (BT) algorithm first suggested by Comiso et al. (1984). In BT algorithm, vertically polarized TBs at 19 and 37 GHz and horizontally polarized TB at 37 GHz are utilized to determine reference TBs (i.e., tie-points) over open water and fully-concentrated ice, which can be used to convert the observed TB to SIC with the following equation:

$$\text{SIC} = \frac{T_B - T_O}{T_I - T_O} \tag{1}$$

where $T_B$ is the satellite-measured TB and $T_O$ and $T_I$ are the empirically determined open water and ice tie-points, respectively. Tie-points in the BT algorithm are updated on a daily basis and acquired separately for the Arctic and the

Antarctic in order to accommodate the variation of TB fields with respect to time and hemisphere (Comiso, 1995). Another well-known algorithm is the NASA Team (NT) algorithm which utilizes horizontally polarized TB at 19 GHz and vertically polarized TBs at 19 and 37 GHz to calculate the polarization ratio (PR) and the spectral gradient ratio (GR), which are used

to determine a set of tie-points to estimate SIC and determine the surface type from a combination of open water, first-year ice, and multi-year ice (Cavalieri et al., 1984). The other is the Arctic Radiation and Turbulence Interaction Study (ARTIST) Sea Ice (ASI) algorithm, which was developed by Kaleschke et al. (2001), in order to exploit the high resolution of the near-90 GHz channels. The ASI algorithm estimates SIC using the tie-points derived from the polarization difference calculated in the near-90 GHz channels. The high sensitivity of the near-90 GHz channels to atmospheric effects is compensated for by

the usage of weather filters, which are applied using the GR thresholds suggested by Gloersen and Cavalieri (1986) and Cavalieri et al. (1995), and by setting SIC to zero in areas where BT SICs are zero (Spreen et al., 2008).

However, discrepancies exist among various PMW SIC records retrieved from different algorithms owing to the different channel combinations, tie-points, and weather filters used in each algorithm (Comiso et al., 1997; Anderson et al., 2007). Due to the lack of reference SIC data with satisfactory temporal and spatial coverages, these disagreements have been

studied mainly through inter-comparison of different PMW SICs and ensemble methods which compare individual SIC products to their averaged value. For instance, Ivanova et al. (2014) reported that different PMW SIC products showed a maximum difference of up to $1.3\times10^6$ km$^2$ in area and $0.6\times10^6$ km$^2$ in extent over the Arctic and larger deviations during the summer due to the differing sensitivity of retrieval algorithms to the presence of melt ponds and the associated emissivity change, as well as a humid atmosphere (Ivanova et al., 2014; Comiso et al., 2017; Horvat et al., 2023). Although these inter-

comparison approaches can provide valuable assessments of the consistency of PMW SIC products from sub-seasonal to climatological timescales, it is noted that there is a limitation to providing a quantitative assessment of PMW SIC products.

In order to make such quantitative assessments, it is essential to have independent SIC data that can be used as a reference. Spaceborne sensors with visible (VIS) to infrared (IR) channels, such as the Moderate Resolution Imaging Spectroradiometer (MODIS) and the sensors onboard the Landsat series, have been used to generate reference SIC due to

their finer spatial resolutions than PMW sensors (Markus et al., 2002, Cavalieri et al., 2006, 2010; Rösel and Kaleschke, 2011; Kern et al., 2022; Tanaka and Lu, 2023; Song and Minnett, 2024). However, validation attempts using VIS/IR-based SIC as a reference have been limited to the usage of a small number of VIS/IR images, with the exception of Kern et al. (2022) which used a relatively large number of Landsat scenes (386 scenes) to generate a reference SIC. The dataset by Kern et al. (2022) is also utilized in this study for validation of the produced Landsat-8 SIC, and the results of the comparison are

presented in Section 3.2. In addition to VIS/IR instruments, SIC observations from synthetic aperture radar (SAR) have also been used for PMW-based SIC validation purposes, but difficulties in obtaining an accurate and automated SIC map from SAR images result in the limited use of SAR images for validation purposes (Anderson et al., 2007; Park et al., 2017; Han and Kim, 2018; Tanaka and Lu, 2023).

Recently, efforts to leverage the advantages of both VIS/IR sensors and PMW sensors for retrieving SIC have been explored

through data merging techniques. Ludwig et al. (2020) used a combination of MODIS and AMSR2 measurements to

construct a high-resolution (1 km) and spatially continuous SIC data over pan-Arctic areas. This approach exploited the benefits of the 1 km resolution MODIS imagery while mitigating its inherent disadvantage of spatial discontinuity due to clouds by introducing the AMSR2 measurements. While the SIC dataset produced by Ludwig et al. (2020) is both high-resolution and covers pan-Arctic areas, due to the retrievals being reliant on the AMSR2 measurements, the product cannot

be considered a fully independent reference data for PMW SIC validation. Therefore, it is still necessary to construct a dataset of Arctic SIC that is fully independent of PMW measurements.

In addition to this, recent applications of deep-learning (DL) models for estimating SIC have shown promising results. Karvonen (2017) trained a multi-layer perceptron (MLP) model using various combinations of PMW signals extracted from AMSR2 and SAR as the training inputs and SIC fields derived from the Finish Meteorological Institute ice charts as the

reference. This MLP model produced improved SIC compared to the high-resolution ASI SIC. However, the data used to train the DL model suggested by Karvonen (2017) were limited to regions around the Baltic Sea and were only acquired during the winter of 2015-2016. Chi et al. (2019) proposed an estimation of Arctic SIC based on MLP model trained with raw AMSR2 TBs as the inputs and SIC derived from seventy-two MODIS images during 2016 as the reference, demonstrating that the DL-based SIC shows better performance than the widely used BT and ASI SICs. Since both studies

used training datasets acquired during a limited time period but showed promising results of the use of DL techniques for SIC production, it is also desirable to construct a data record for reference SIC data with satisfactory temporal and spatial coverages.

Therefore, this study aims to construct a reference SIC dataset of satisfactory spatiotemporal extent, to allow for validation of various SIC products over pan-Arctic areas as well as to be used for DL training. To do this, a total of 14,297 Landsat-8

images over the three years (2020-2022) were translated into SIC maps in a 6.25 km polar stereographic grid and catalogued into a region of the Arctic Ocean.

The remaining sections of this paper are organized as follows: Section 2 provides a detailed description of the Landsat-8 dataset, the land, sea ice region, coastal area masks, and the reference datasets used to evaluate the Landsat-8 SIC in this study. Section 3 describes the processing pipeline of a Landsat-8 image into a SIC dataset along with the uncertainty

estimation. The resultant SIC product and its uncertainty are shown in Section 4. Possible errors in Landsat-8 SIC, evaluation of Landsat-8 SIC using existing SIC from ice charts, evaluation of Landsat-8 SIC over melt ponds, and qualitative assessment of two PMW SIC products using SIC from Landsat-8 as a reference are discussed in Section 5. Section 6 provides the data availability statement, and Section 7 gives the summary and conclusion of this research.


## 2 Used Data

### 2.1 Landsat-8 OLI-TIRS Collection 2 Level 1 Products

In this study, reflectivities measured by the Operational Land Imager (OLI) onboard Landsat-8, which is a polar orbiting satellite with orbit inclination of 98.2° and a repeat cycle of 16 days (Zanter, 2019), were used to retrieve SICs over pan-Arctic areas. The OLI sensor has a swath width of 185 km, measuring radiances at eight bands from VIS to short-wave IR (SWIR) with a spatial resolution of 30 m. It should be noted that areas with a latitude higher than 82°N in the northern hemisphere are not measured by Landsat-8 (i.e., the hatched area in Fig. 1) due to the orbit inclination of Landsat-8 and the relatively narrow swath width of the OLI. The Landsat-8 Collection 2 Level 1 product used in this study contains eleven spectral band images (nine bands from the OLI and two bands from the Thermal Infrared Sensor) provided in GeoTIFF format, two quality assessment bands containing masking information for clouds, cloud shadows, cirrus, fill values, and radiometric saturation. To calculate the SIC, the OLI-measured reflectivities at near-infrared (NIR) band 5 and SWIR band 6 (used in the Normalized Difference Snow Index) were used in this study. It is worth noting that the methods developed in this study (described in Section 3) utilize the NIR and SWIR bands for SIC retrieval and are therefore applicable to a wider range of high-resolution sensors that observe at similar bands, including the Multi-Spectral Instrument (MSI) onboard Sentinel-2. However, due to the more robust cloud mask performance of the Landsat-8 product, in this study, the Landsat-8 Collection 2 Level 1 product was selected to be used for the production of reference SIC data (Zhu et al., 2015; Tarrio et al., 2020).

For the period of Jan. 2020 – Dec. 2022, Landsat-8 Collection 2 Level 1 product and the corresponding true-color images were downloaded from the United States Geological Survey Earth Explorer (https://earthexplorer.usgs.gov/). To circumvent the influences of cloud contamination and low solar elevation angle on SIC calculation, which hampers accurate classification of ice and open water, only Landsat-8 images with less than 10% cloud cover (based on fractional cloud masked area from the quality assessment band of Landsat-8) during daytime (solar elevation higher than 15°) were collected. While the threshold of 0% cloud cover would ensure the acquisition of the least cloudy scenes, this also results in the loss of a considerable number of Landsat-8 scenes that contain clear-sky portions (see Fig. S1 and Table S1 in the supplements for the number of available Landsat-8 scenes subject to different threshold values of cloud cover). Therefore, the threshold value for cloud cover was relaxed to 10% during the acquisition of Landsat-8 images. Since VIS measurements are not available during polar night-time, the Landsat-8 data between early December to January were not collected. A total of 15,286 images were collected and sorted into twelve regions of the pan-Arctic area for the calculation of SIC. In the case of a Landsat-8 image that observed across more than one region, the image was sampled repeatedly for each region. Footprints of the collected Landsat-8 images are displayed in Fig. 1 and the number and the temporal availability of the collected images for each area are listed in Table 1.

| Region | Baffin Bay | Barents Sea | Beaufort Sea | Bering Sea | Canadian A. | Central Arctic |
|--------|------------|-------------|--------------|------------|-------------|----------------|
| Count | 2476 | 672 | 451 | 699 | 3174 | 2343 |
| 2020 | Jan. 13-Nov. 30 | Feb. 27-Oct. 7 | Mar. 6-Sep. 17 | Jan. 19-Dec. 3 | Feb. 29-Oct. 2 | Mar. 23-Sep. 2 |
| 2021 | Feb. 16-Nov. 26 | Feb. 22-Oct. 5 | Mar. 7-Sep. 30 | Jan. 13-Dec. 3 | Mar. 1-Sep. 28 | Mar. 24-Sep. 11 |
| 2022 | Jan. 25-Dec. 6 | Mar. 4-Sep. 29 | Mar. 4-Oct. 4 | Jan. 24-Nov. 16 | Mar. 3-Oct. 10 | Mar. 25-Sep. 13 |
| Region | Chukchi Sea | E. Greenland | E. Siberian | Hudson Bay | Kara Sea | Laptev Sea |
| Count | 427 | 1468 | 546 | 1485 | 899 | 646 |
| 2020 | Feb. 27-Oct. 10 | Feb. 18-Oct. 31 | Mar. 4-Sep. 21 | Jan. 13-Nov. 14 | Mar. 6-Sep. 16 | Mar. 10-Sep. 27 |
| 2021 | Feb. 25-Oct. 11 | Feb. 25-Oct. 27 | Mar. 7-Oct. 2 | Jan. 17-Nov. 6 | Mar. 7-Sep. 17 | Mar. 15-Sep. 13 |
| 2022 | Mar. 6-Oct. 14 | Mar. 4-Oct. 30 | Mar. 10-Sep. 7 | Jan. 18-Oct. 29 | Mar. 5- Sep. 29 | Mar. 12-Sep. 6 |

**Table 1: The number of Landsat-8 images collected in this study and the available period for each region of the pan-Arctic areas**

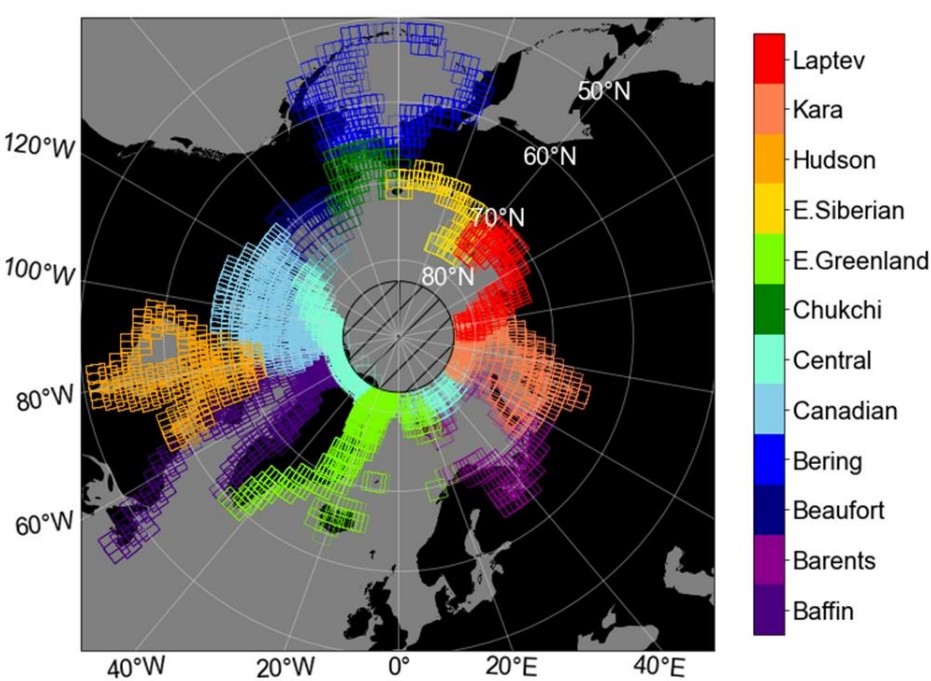

**Figure 1: Footprints of the collected Landsat-8 images over each region of the pan-Arctic areas during the period of Jan. 2020 –**
**Dec. 2022. The hatched region denotes the areas unmeasured by Landsat-8 due to its orbital inclination (i.e., pole hole). The regions of the pan-Arctic areas were distinguished using the region mask provided by Meier and Stewart (2023). The map projection is NSIDC Sea Ice Polar Stereographic North (EPSG: 3413) and the map was plotted using Python.**

## 2.2 Land and Sea Ice Region Masks

Regions of the Arctic Ocean were distinguished using the National Snow and Ice Data Center (NSIDC) 'Arctic and Antarctic Regional Masks for Sea Ice and Related Data Products, Version 1' data (Meier and Stewart, 2023), which divides the Arctic Ocean into nineteen different regions with 6.25, 12.5, and 25 km resolution polar stereographic (PSR) grids. In addition, this product provides surface masking information to differentiate between ocean areas and non-ocean areas such as land, freshwater, land ice, ice shelf, and disconnected ocean. The flag values for the Arctic Ocean regions and the different surface

types can be found in the product user guide (Meier and Stewart, 2023). Amongst the regions, twelve regions (i.e., Baffin Bay and Labrador Seas, Barents Sea, Beaufort Sea, Bering Sea, Canadian Archipelago, Central Arctic, Chukchi Sea, East Greenland Sea, East Siberian Sea, Hudson Bay, Kara Sea, and Laptev Sea; Fig. 2) were selected to generate Landsat-8 based SICs, because the above twelve regions have climatologically meaningful sea ice extent.


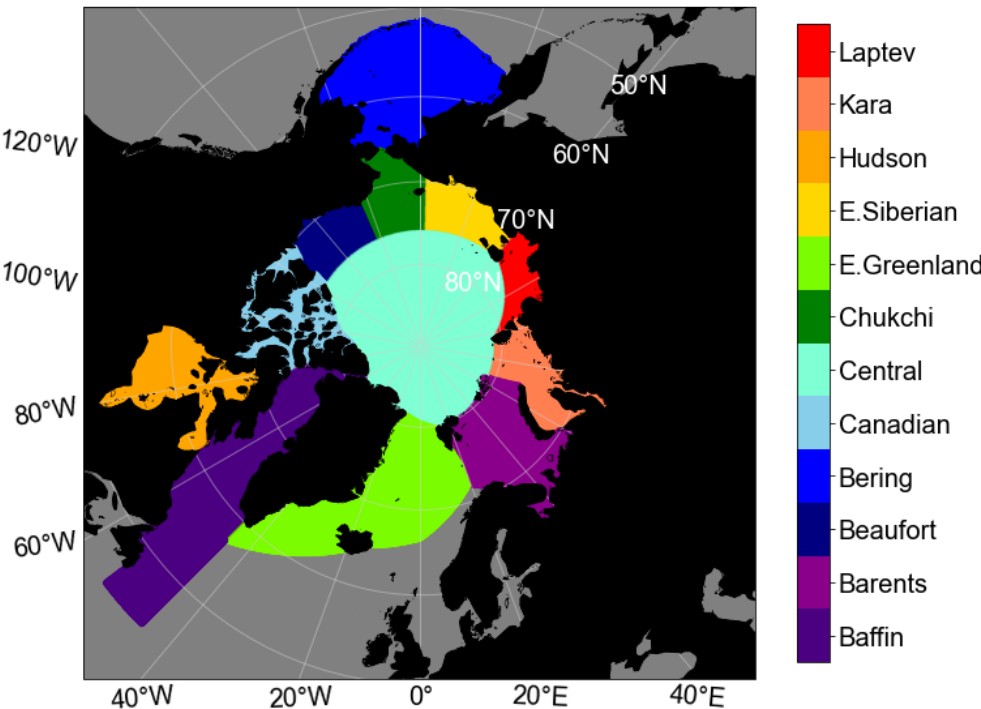

**Figure 2: Geographic distribution of the designated regions of the Arctic Ocean, based on NSIDC Sea Ice Region Mask data (Meier and Stewart, 2023). The map projection is NSIDC Sea Ice Polar Stereographic North (EPSG: 3413) and the map was plotted using Python.**


### 2.3 Ice/Water Classified Landsat-8 Images

The performance of ice and open water classification (later described in Section 3.2), in this study, was evaluated using ice-water classification data from the 'Land surface type over water from supervised classification of surface broadband albedo estimates' (Kern, 2021; Kern et al., 2022). This dataset contains ice/water classification estimates using broadband albedo values from the Landsat series (i.e., bandwidth weighted mean albedo from Landsat-8-measured reflectivities at bands 3, 4, and 5), where each pixel in a scene is classified into open water, thin or bare ice, and thick or snow-covered ice based on supervised classification. In our study, the two ice categories (i.e., one for thin or bare ice and the other for thick or snow-covered ice) were considered as a same ice category due to the higher ambiguities in the discrimination among different ice types relative to the discrimination between ice and open water (Kern et al., 2022). In order to evaluate the classification method suggested by our study we processed Landsat-8 reflectance from six clear-sky scenes that Kern (2021) had classified, and then compared results. The result of the comparison is presented in Section 3.2 and the location and time of the Landsat-8 scenes that were used in the evaluation are provided in the supplements Fig. S2 and Table S2.

### 2.4 Ice Chart Data

Ice charts provide SIC intervals over the Arctic obtained by manual interpretation of satellite images from various sensors such as SAR, MODIS, and the Advanced Very High Resolution Radiometer (AVHRR). In this study, operational ice charts from Norwegian Meteorological Institute (MET Norway), which provide SIC maps in PSR grid with nominal resolution of 1 km, were used to evaluate the performance of the produced Landsat-8 SIC. Each grid in the ice chart contains the classified six SIC values (5, 20, 50, 75, 95, and 100%), which represent the ice concentration intervals defined by the World Meteorological Organization (WMO) (Table A1). The ice charts are provided on a daily basis and cover the spatial domain of approximately 80°W-80°E, 60°N-85°N, which overlap with the regions of Barents Sea, Central Arctic, East Greenland Sea, and Kara Sea defined in Section 2.2. It is noted that SIC in ice charts are based on the interpretation of multiple satellite imageries by ice analysts, and therefore contain high uncertainties, which are reflected by the wide ice concentration intervals designated for each of the six SIC values (Table A1). Even with such high uncertainties, SIC from ice charts have been widely selected as reference data in SIC product validation studies, because they can be used to provide quantitative information about the observed ice coverage (Agnew and Howell, 2010; Ivanova et al., 2015; Karvonen, 2017).

In this study, two-years (2021 and 2022) of ice charts were collected among which 222 ice charts that have spatial overlap with the coverage of Landsat-8 SIC and have time difference of less than 1 hour with the Landsat-8 scene were used for evaluation of the produced Landsat-8 SIC (see Table S3 in the supplements for the list of ice charts and Landsat-8 files used in the evaluation of the produced Landsat-8 SIC).

**2.5 Melt Pond Fraction Data**

Melt ponds are formed from the surface melting of sea ice and are known to exist in preponderance over the Arctic during the melting season (Untersteiner, 1961; Fetterer and Untersteiner, 1998; Rösel et al., 2012). In the VIS/IR ranges, melt ponds typically exhibit lower spectral reflectivities relative to dry sea ice (Perovich, 1996; Malinka et al., 2018), and therefore may introduce errors in SIC estimated from VIS/IR observations because the optical characteristics of melt ponds may not be differentiated from those of open ocean. In order to test the sensitivity of Landsat-8 SICs to the existence of melt ponds, in this study, a melt pond fraction (i.e., the fractional areal coverage of melt ponds over sea ice; MPF) dataset estimated from clear-sky Sentinel-2 satellite imagery was introduced (Niehaus and Spreen, 2022; Niehaus et al., 2023). This dataset also contains an open water mask (OW mask), which is a binary classification mask of each pixel in a Sentinel-2 scene into ice and open water. This dataset is available from 2017 to 2021 for the Arctic melting season (i.e., June, July, and August). In this study, each MPF dataset was tested for spatiotemporal overlap (time difference of less than 3 hours) with the coverage of Landsat-8 SIC. The total of six MPF datasets were found to be overlapping with the coverage of Landsat-8 SIC and thus available for use in the evaluation. The list of available MPF datasets and the corresponding Landsat-8 scenes can be seen in Table S4 of the supplements.

**3 Method**

Figure 3 shows the processing pipeline of a Level 1 Landsat-8 image into a SIC product based on 6.25 km resolution PSR grid. Details of each process are explained as the following sub-sections.

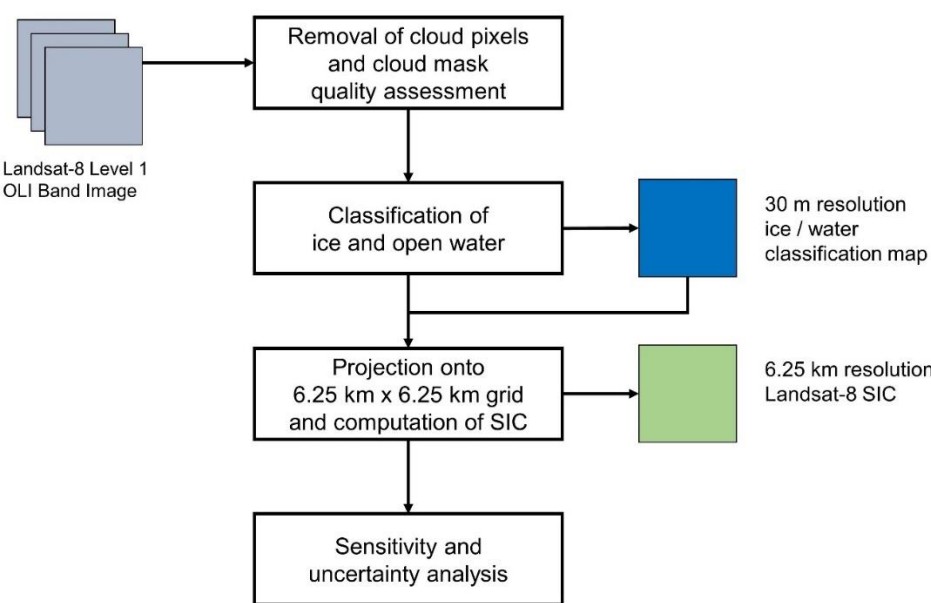

**Figure 3: Processing pipeline of Level 1 Landsat-8 OLI images into SICs with 6.25 km resolution.**

### 3.1 Removal of Cloud Pixels and Cloud Mask Quality Assessment

Satellite observations of surface properties from the VIS and NIR channels are hindered by the presence of clouds. Therefore,
it is important to filter the presence of clouds prior to the SIC data production. In this study, clouds and cloud shadows within each Landsat-8 scene were masked using the masking array constructed from the quality assessment band of each Landsat-8 Level 1 product, which is generated by the C Function of Mask (CFMask) (Zhu and Woodcock, 2012). CFMask is a cloud detection algorithm that provides masking information for clouds, cloud shadows, and cirrus. Confidence scores are also given in three levels (i.e., low, medium, and high) for clouds and two levels (i.e., low and high) for cirrus. Confidence
score for cloud shadows is not provided because cloud shadows are only derived from high confidence cloud pixels by using the geometric relationship between the position of the sun and high confidence cloud pixels (Zhu and Woodcock, 2012). Although the application of the lowest confidence scores in the removal of clouds and cirrus would ensure the lowest rate of false negatives (FN; cloud pixels that are mistaken as clear pixels) in cloud detection, the use of the lowest confidence scores also result in the removal of a considerable number of sea ice pixels under clear sky (Foga et al, 2017). Therefore, it is
important to select proper confidence scores to retain as many clear sky sea ice pixels as possible while minimizing the number of FN cases. In this study, pixels with medium and high confidence scores for clouds and for cirrus, respectively, were discarded prior to Landsat-8 SIC production to avoid cloud and cirrus contamination. In addition, as suggested in Foga et al. (2017), dilated cloud pixels, which are clear pixels completely surrounded by cloud pixels, were also masked to prevent contamination by cloud edges where cloud detection uncertainty is high.

It is important to note that CFMask over ice-covered surface types has lower accuracy than other surface types (Foga et al., 2017; Qiu et al., 2019). Therefore, an additional step for cloud mask quality assessment is designated in this study. In this step, a visual inspection was performed by comparing the cloud mask array, which is constructed by masking cloud, cirrus, cloud shadow, and dilated cloud pixels, from each Landsat-8 image with the corresponding true-color image to identify the cases of FN, false positive (FP; clear pixels that are mistaken as cloud pixels), true negative (TN; clear pixels correctly
detected as clear pixels), and true positive (TP; cloud pixels correctly detected as cloud pixels) pixels in the Landsat-8 image. From this additional step, Landsat-8 images were sorted into four categories depending on the assessed quality of cloud masking. Images with the existence of FN cloud pixels in the cloud mask array, which indicate the underestimated cloud cover, were labelled as Category 1 (C1). Images dominated by FP cloud pixels, which occur in cases of the overestimated cloud cover, were tagged as C2. Images dominated by TP cloud pixels, which correspond to correctly estimated cloud cover
for cloudy sky, were labelled as C3. Images dominated by TN cloud pixels, which correctly estimate clear sky, were labelled as C4. For images under C2 (i.e., overestimated cloud coverages with medium confidence scores for clouds and high confidence scores for cirrus), the cloud mask array was regenerated with a higher confidence score (high confidence clouds and cirrus) and visually inspected against the true-color image to determine the adequacy of the higher confidence score cloud mask as follows: If any FN cloud pixels were present in the higher confidence cloud mask, the original confidence

score (i.e., medium for clouds and high for cirrus) was used to mask the clouds. Further details of the visual screening step are provided in Appendix B.

In this study, for Landsat-8 images that were labelled as C2, C3, and C4, Landsat-8 pixels that remain after the application of CFMask were assumed to be clear sky pixels (i.e., "clear pixel assumption"). However, for Landsat-8 images labelled as C1, the "clear pixel assumption" is not valid because C1 category underestimates clouds by CFMask according to the visual

inspection step, which implies that the associated error due to the underestimated cloud cover in SIC calculation is expected. Therefore, possible error from the presence of unmasked cloud pixels in C1 is further evaluated in Section 5.1. The number of Landsat-8 images under the four categories over the twelve regions is provided in Table 2, and the assessed cloud mask quality (i.e., C1, C2, C3, and C4) for each Landsat-8 image is provided in the variable under the name 'cloud_contamination_category' in the produced Landsat-8 SIC dataset in order to allow for quality control of the data in its

usage.

| Category | C1[a] | C2[b] | C3[c] | C4[d] |
|---|---|---|---|---|
| Baffin Bay | 826 | 80 | 907 | 663 |
| Barents Sea | 271 | 10 | 265 | 126 |
| Beaufort Sea | 215 | 27 | 111 | 98 |
| Bering Sea | 209 | 24 | 264 | 202 |
| Canadian A. | 1,573 | 176 | 854 | 571 |
| Central Arctic | 1,165 | 42 | 705 | 431 |
| Chukchi Sea | 154 | 30 | 134 | 109 |
| E. Greenland | 767 | 29 | 369 | 303 |
| E. Siberian | 230 | 36 | 145 | 135 |
| Hudson Bay | 619 | 116 | 351 | 399 |
| Kara Sea | 490 | 34 | 245 | 130 |
| Laptev Sea | 328 | 23 | 165 | 130 |

[a]underestimated cloud cover
[b]overestimated cloud cover
[c]correctly estimated cloud cover for cloudy sky
[d]correctly estimated cloud cover for clear sky

**Table 2: The number of Landsat-8 images for the four cloud mask categories (i.e., C1: underestimated cloud cover, C2: overestimated cloud cover, C3: correctly estimated cloud cover for cloudy sky, and C4: correctly estimated cloud cover for clear sky) over the twelve regions of the Arctic Ocean during the periods of Jan. 2020 – Dec. 2022.**


## 3.2 Ice and Open Water Classification

Classification of a Landsat-8 pixel as ice or open water was performed by applying thresholds to the top-of-atmosphere (TOA) reflectivity at band 5 (NIR) and the normalized difference snow index (NDSI). To do this, first, the reflectivity of a Landsat-8 pixel, which is stored as a 16-bit digital number in the Landsat-8 Collection 2 Level 1 dataset, was scaled to TOA reflectivity using the following equation (Zanter, 2019):

$$\rho_\lambda = \frac{M_\rho Q_{DN} + A_\rho}{\sin(\theta_{SE})} \tag{2}$$

where $M_\rho$, and $A_\rho$ are the multiplicative and additive scale factors, $\theta_{SE}$ is the solar elevation angle, and $Q_{DN}$ is the TOA reflectivity of the Landsat-8 pixel in 16-bit digital number format.

Then, the NDSI was calculated from the scaled reflectivities as follows:

$$NDSI = \frac{\rho_5 - \rho_6}{\rho_5 + \rho_6} \tag{3}$$

where $\rho_5$ and $\rho_6$ are the TOA reflectivities at bands 5 (NIR) and 6 (SWIR) of the OLI sensor, respectively.

The steps for differentiating ice and open water pixels and for removing possible cloud pixels are shown in Fig. 4. The first is the $\rho_5$ criterion in order to detect open water pixels, which has lower reflectivity at band 5 compared to that over ice or cloud pixels. The next step is the NDSI criterion in order for detecting ice pixels, which has higher NDSI than cloud pixels, due to higher reflectivity of ice at band 5 and lower reflectivity of ice at band 6, compared to the cloud reflectivities (Hall et al., 1995; Riggs et al., 1996, 1999). The NDSI criterion for the separation of ice and cloud pixels was kept in order to reinforce cloud removal process in addition to CFMask explained in Section 3.1. In this study, the thresholds for $\rho_5$ and NDSI were selected as 0.08 and 0.45, respectively (Liu et al., 2016; Tanaka and Lu, 2023).

As mentioned in Section 3.1, the "clear pixel assumption" was applied during the classification of Landsat-8 images labelled C2, C3, and C4. Accordingly, the performance of classification based on $\rho_5$ and NDSI with the selected thresholds was evaluated over clear sky pixels using the surface classification data from Landsat-8 images (Kern, 2021) mentioned in Section 2.3 as reference data. The values of $\rho_5$ and NDSI were collected separately over open water and ice pixels in the reference data and classification over the collected pixels was performed following the procedure in Fig. 4. From the distributions of ice and open water pixels in the two-dimensional histogram between NDSI (x-axis) and $\rho_5$ (y-axis) in Fig. C1, it can be seen that ice and open water are well differentiated by the selected threshold values of $\rho_5$ and NDSI, respectively (Fig. C1). In addition, for quantitative assessment of the performance of ice and open water classification, the recall was computed for the open water and ice categories using the classification result summarized in Table 3 and Eq 4.

$$RC_{X \text{ as } X} = \frac{N_{X \text{ as } X}}{N_{X \text{ as } X} + N_{X \text{ as } \sim X}} \tag{4}$$

where $N_{X \text{ as } X}$ and $N_{X \text{ as } \sim X}$ are the number of pixels in category X classified as X (TP) and the number of pixels in category X classified as not X (FN), respectively. With the designated thresholds the recall was found to be 98.94% for water and 97.67% for ice. FN classifications of ice into open water can cause negatively biased SIC. The bias due to such classification error was estimated to be 2.33% from the percentage of the number of ice pixels that were classified as open water in Table 3. Conversely, FN classification of open water into ice can cause positively biased SIC, which was estimated to be 1.06% from the value in Table 3. Misclassification of ice or open water pixels into cloud pixels from the application of the NDSI threshold rarely occurred for both ice and open water categories. Thus, it can be concluded that the classification method used in this study is accurate over clear sky pixels. Furthermore, the error from ice/water classification over clear sky is within the uncertainty range of Landsat-8 SIC which is discussed in Section 4.3.

This classification result may not be applicable for Landsat-8 images tagged C1 (i.e., underestimated cloud cover), because as mentioned in Section 3.1, such images do not consist solely of clear sky pixels, but contain undetected cloud pixels by CFMask. Therefore, for Landsat-8 images labelled C1, in order to understand possible errors in SIC calculation from the designated classification method, it is necessary to evaluate the performance of classification over the undetected cloud pixels. This is discussed further in Section 5.1.

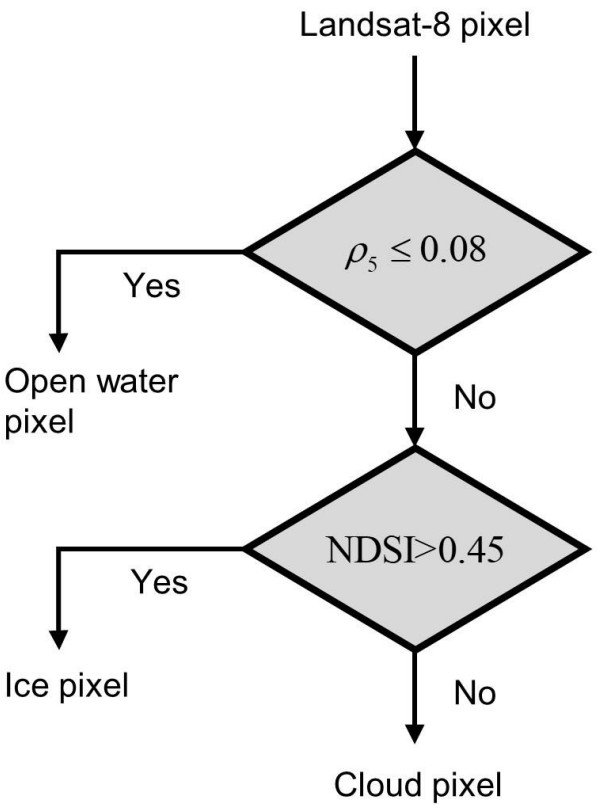

**Figure 4: The process for separating ice, open water, and possible unmasked clouds using $\rho_5$ and NDSI criterion, where $\rho_5$ and NDSI are the TOA reflectivity at band 5 of the OLI sensor and the Normalized Difference Snow Index, respectively.**

| Classified / Reference | Open water | Ice | Cloud |
|---|---|---|---|
| Open water | 13,271,877 (98.94%) | 141,582 (1.06%) | 19 (0.00%) |
| Ice | 747,481 (2.33%) | 31,353,954 (97.67%) | 336 (0.00%) |

**Table 3: The number of classified pixels for open water, ice, and cloud from the suggested method and surface classification reference data (Kern, 2021). The original categories in the reference data are shown in the rows, and the classified categories from the method in Fig. 4 are shown in the columns. The values inside the parentheses indicate the percentage of pixels from the original category that are classified into open water, ice, and cloud.**

### 3.3 Projection and Computation of SIC

After the ice and open water classification for the selected Landsat-8 pixels, the classified pixels were projected onto the target grid system of the NSIDC Polar Stereographic grid with 6.25 km resolution. The number of ice and open water pixels within each 6.25 km × 6.25 km grid cell was used to compute SIC for the grid cell according to,

$$SIC = \frac{N_{ice}}{N_{ice} + N_{water}} \times 100\,(\%) \tag{5}$$

where $N_{ice}$ and $N_{water}$ are the number of Landsat-8 pixels classified as ice and water within each 6.25 km × 6.25 km grid cell, respectively. It is noted that some of the grid cells with 6.25 km resolution are not entirely filled by Landsat-8 pixels at the edges of a Landsat-8 image and/or near cloud masked regions. In this study, this kind of grid cell is referred to as 'partially-covered grid cell'. Therefore, SIC computed in such a grid cell is unlikely to be representative of the actual ice coverage over the area covered by the grid cell. To avoid this caveat, a minimum threshold in the number of Landsat-8 pixels for a single 6.25 km × 6.25 km grid cell ($N_{critical}$) was applied prior to the computation of SIC. In this study, a specific value of $N_{critical}$ was introduced as the minimum threshold, which is discussed in the following subsection.

### 3.4 Sensitivity Test and Uncertainty Analysis

The sensitivity of Landsat-8 SIC to the prescribed thresholds of $\rho_5$ and NDSI was investigated for each cloud contamination category. In doing so, for each of the four cloud contamination categories (i.e., C1, C2, C3, and C4), ten scenes were randomly sampled over all twelve regions (Fig. 2), and thus 120 scenes were used per each cloud contamination category for sensitivity test (see Table S5 in the supplements for the full list of scenes used for sensitivity test). SIC over the selected scenes were calculated using Eq. 5 based on classification results with NDSI and $\rho_5$ thresholds perturbed by their

uncertainties. Values of 0.015 and 0.016 were assigned as the uncertainties of $\rho_5$ and $\rho_6$, respectively, followed by Pinto et al. (2020) which provides the root mean squared differences of the Landsat-8 TOA reflectivities and in situ observed

reflectivities at bands 5 and 6. The uncertainty of NDSI was calculated using the Gaussian error propagation method, which can be written for NDSI as:

$$\sigma_{NDSI}^2 = \left(\frac{\partial NDSI}{\partial \rho_5}\right)^2 \sigma_{\rho_5}^2 + \left(\frac{\partial NDSI}{\partial \rho_6}\right)^2 \sigma_{\rho_6}^2 \qquad (6)$$

where $\sigma_{\rho_5}$ and $\sigma_{\rho_6}$ are the uncertainties of Landsat-8 TOA reflectivities at bands 5 and 6, respectively. Substituting Eq. 3 for NDSI in Eq. 6, the analytical form of the uncertainty in NDSI can be expressed as the following:

$$\sigma_{NDSI}^2 = \frac{4\rho_6^2}{(\rho_5 + \rho_6)^4}\sigma_{\rho_5}^2 + \frac{4\rho_5^2}{(\rho_5 + \rho_6)^4}\sigma_{\rho_6}^2 \qquad (7)$$

From Eq. 7 with $\sigma_{\rho_5} = 0.015$ and $\sigma_{\rho_6} = 0.016$, a value of 0.05 was assigned as the uncertainty of NDSI which is the median value of $\sigma_{NDSI}$ computed over 480 randomly selected Landsat-8 scenes. For the four cloud contamination categories, mean values of SICs calculated with the perturbed thresholds of 0.45±0.05 and 0.08±0.015 for the NDSI and $\rho_5$, respectively, are provided in Fig. 5. With the perturbation of ±0.015 for $\rho_5$ threshold, mean SICs from C1, C2, C3, and C4 vary by

$\mp$0.641%, $\mp$0.495%, $\mp$0.665%, and $\mp$0.402%, respectively (blue lines in Fig. 5). With perturbation of ±0.05 for NDSI threshold, mean SIC from C1, C2, C3, and C4 varied by $\mp$0.111%, $\mp$0.002%, $\mp$0.007%, and $\mp$0.002%, respectively (red lines in Fig. 5). The calculated SICs are more sensitive to the $\rho_5$ threshold relative to the NDSI threshold because the $\rho_5$ threshold is responsible for separating open water and ice. It is noted that sensitivity of SICs to the NDSI threshold is two-orders of magnitude higher for scenes labelled C1 than for C2, C3, and C4. The very low sensitivity of SICs to the NDSI

threshold for scenes labelled C2, C3, and C4 infers that cloud pixels in such scenes had been successfully masked by CFMask prior to the ice/water classification described in Section 3.2. However, relatively higher sensitivity of SICs to the NDSI threshold for scenes under C1 infers that undetected cloud pixels had remained after the application of CFMask and that such cloud pixels had been further removed by the NDSI threshold.

Gaussian error propagation was also used to estimate the uncertainty of Landsat-8 SIC according to:

$$\sigma_{SIC}^2 = \left(\frac{\partial SIC}{\partial NDSI}\right)^2 \sigma_{NDSI}^2 + \left(\frac{\partial SIC}{\partial \rho_5}\right)^2 \sigma_{\rho_5}^2 \quad (\%) \qquad (8)$$

where $\sigma_x$ and $\frac{\partial SIC}{\partial x}$ are the uncertainty of $x$ and the sensitivity of SIC to $x$, respectively. The sensitivities for the two variables (i.e., $\rho_5$ and NDSI) were computed numerically from the mean SIC variation observed in sensitivity test (see Tables S6, S7, S8, and S9 in the supplementary for the computed values of sensitivity). In addition, in order to check the relative

contribution of each variable to the overall uncertainty in SIC, a contribution factor ($CF_x$) was defined and calculated for the

two variables as the following:

$$CF_x = \frac{\left(\dfrac{\partial SIC}{\partial x}\right)^2 \sigma_x^2}{\sigma_{SIC}^2} \times 100 \ (\%) \tag{9}$$

The estimated uncertainty of Landsat-8 SIC ($\sigma_{SIC}$) produced in this study was less than 1% in average for all four cloud contamination categories and the $\rho_5$ threshold contributes to about 99% of the uncertainty for C2, C3, and C4 and about 97% of the uncertainty for C1 in SIC calculation. Further discussion of the uncertainty of Landsat-8 SIC is handled in Section 4.3.

As mentioned in Section 3.3, SIC computed from partially-covered grid cells may not be representative of actual ice coverage over the entire grid cell and the corresponding uncertainty of SIC estimates in such grid cell can be as large as the fraction of the uncovered areas. In order to circumvent such a problem, in this study, $N_{critical}$ was determined as $0.99 \times N_{max}$ where $N_{max}$ is the maximum number of Landsat-8 pixels within a 6.25 km × 6.25 km grid cell.

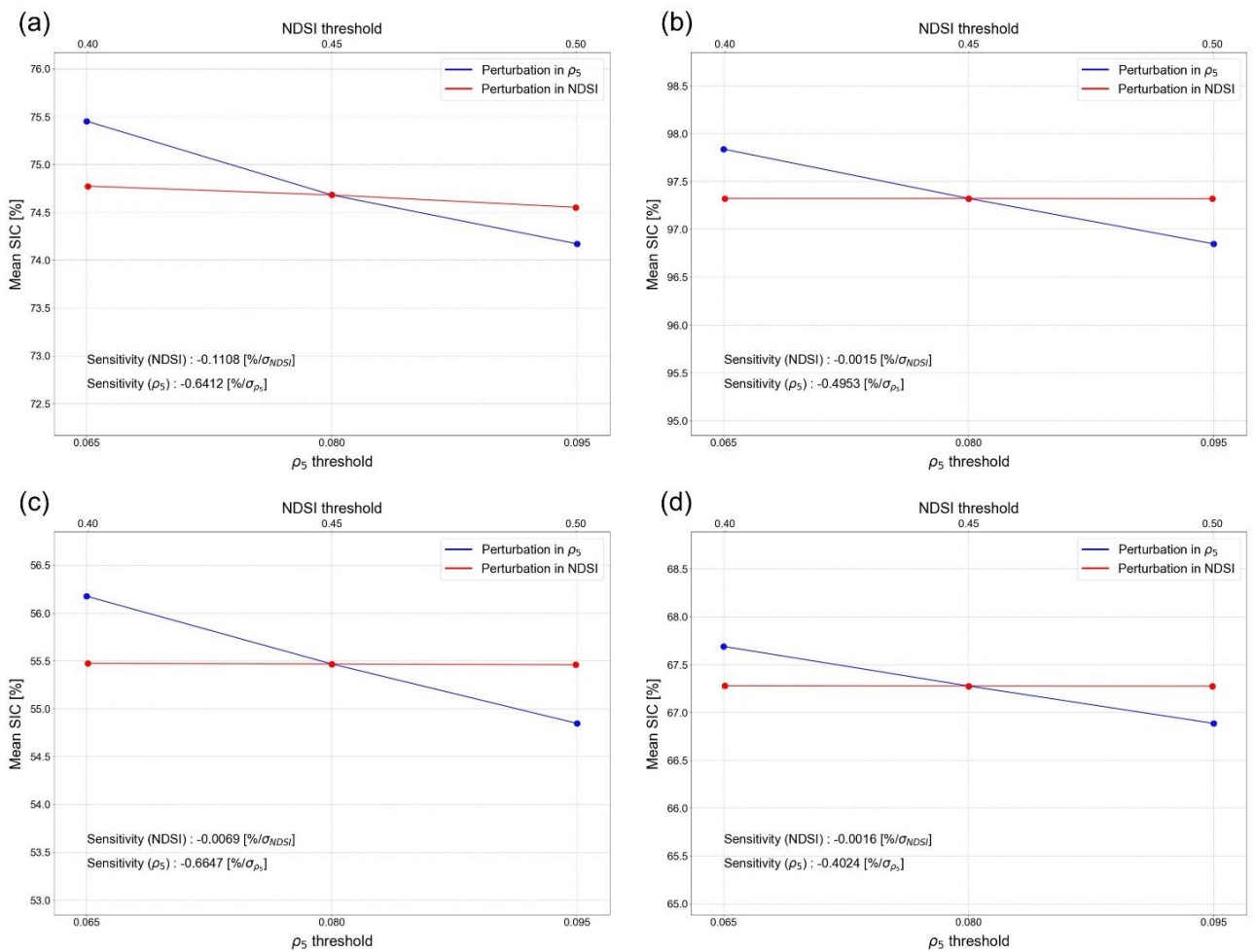

**Figure 5: Mean of Landsat-8 SIC values for (a) C1 (i.e., underestimated cloud cover), (b) C2 (i.e., overestimated cloud cover), (c) C3 (i.e., correctly estimated cloud cover for cloudy sky), and (d) C4 (i.e., correctly estimated cloud cover for clear sky) derived from the selected scenes under perturbed thresholds for NDSI (red) and $\rho_5$ (blue), where $\rho_5$ and NDSI are the TOA reflectivity at band 5 of the OLI sensor and the Normalized Difference Snow Index, respectively.**

## 3.5 Application of Land and Region Masks

In order to circumvent potential contamination of land signals, in this study, SIC pixels generated over non-ocean regions were masked using the surface mask described in Section 2.2. The region mask was applied in addition to the surface mask to obtain SIC products catalogued into the 12 regions. If all SIC pixels in a Landsat-8 scene were masked by the combination of land, region, and cloud masks, the scene was removed from the SIC dataset.

# 4 Result

## 4.1 Landsat-8 SIC Dataset

Out of 15,286 Landsat-8 Level 1 images collected over pan-Arctic areas during the study period, the number of Landsat-8 images used for calculating SICs for the categories of C1, C2, C3, and C4 were 6,336 (41.4%), 549 (3.6%), 4,389 (28.1%), and 3,123 (20.4%), respectively. The remaining 989 (6.5%) images were removed from the combination of surface, region, and cloud masks. For each of the twelve regions, the number of Landsat-8 scenes generated into Landsat-8 SIC ($N_{scene}$), and the number of produced Landsat-8 SIC pixels ($N_{pixel}$) for each cloud contamination categories during the study period are

shown in Fig. 6 along with the mean and standard deviation of SIC (see Table S10 in the supplementary for values). The total number of Landsat-8 SIC pixels produced during the study period was 2,934,399.

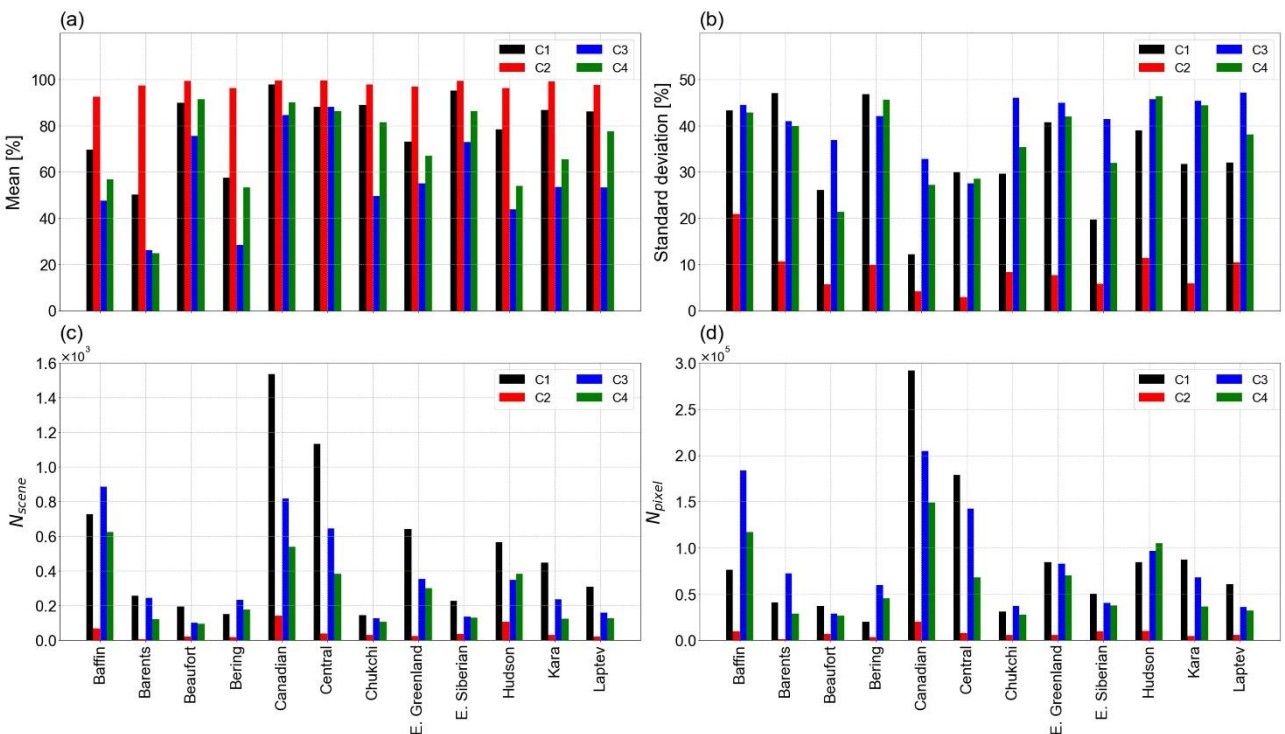

**Figure 6: (a) The mean SIC, (b) the standard deviation of SIC, (c) the number of Landsat-8 scenes used for SIC production ($N_{scene}$),**
**and (d) the number of Landsat-8 SIC pixels produced ($N_{pixel}$) over the twelve regions. The black, red, blue, and green bars indicate values for categories C1 (i.e., underestimated cloud cover), C2 (i.e., overestimated cloud cover), C3 (i.e., correctly estimated cloud cover for cloudy sky), and C4 (i.e., correctly estimated cloud cover for clear sky), respectively.**

## 4.2 Qualitative Evaluation for Landsat-8 SIC Under Four Cloud Contamination Categories

Figure 7 shows the Landsat-8 true-color image (first column of Fig. 7), classification map of ice, open water, and the removed cloud pixels (second column of Fig. 7), and Landsat-8 SIC in 6.25 km resolution (third column of Fig. 7) for the four selected cases. Ice and open water pixels, which were differentiated following the methods explained in Section 3.2, are shown as the white and blue pixels, respectively. Cloud pixels removed in both CFMask and the NDSI criterion are shown as the cyan pixels. Cloud pixels removed from CFMask but undetected from the NDSI criterion are shown as the purple pixels.

Cloud pixels removed from the NDSI criterion but undetected in CFMask are shown as the red pixels. SICs were only estimated over grid cells that satisfy $N > N_{critical}$, therefore grid cells with more than 1% of its area covered with cloud pixels or grid cells located near the edges of a Landsat-8 scene have no SIC values. In addition, areas close to the coastline (within 6.25 km) are masked in the Landsat-8 SIC maps presented in Fig. 7.

The first case is an example of the underestimated cloud cover (i.e., C1) on March 13, 2022 over the Kara Sea (first row of

Fig. 7) where cloud pixels observed in the lower left area of Fig. 7a were not removed by CFMask (cyan and purple pixels in Fig. 7b). However, for this particular scene, most of such undetected cloud pixels were removed from application of the NDSI criterion (red pixels in Fig. 7b) and therefore the produced SIC was estimated only over clear sky area (Fig. 7c). The second case is an example of the overestimated cloud cover (i.e., C2) on March 17, 2021 over the Barents Sea (second row of Fig. 7) where FP cloud pixels are densely distributed in the upper left area of Fig. 7e. It is shown that SICs were not

estimated for grid cells with such wrongly-masked pixels (Fig. 7f). The third is an example of correctly estimated cloud cover for cloudy sky (i.e., C3) on June 26, 2022 over the Kara Sea (third row of Fig. 7) where the position of cloud pixels removed from CFMask (cyan and purple pixels in Fig. 7h) coincide well with the location of cloud presented in the true-color image (Fig. 7g). The fourth case is an example of correctly estimated cloud cover for clear sky (i.e., C4) on June 15, 2022 over the Beaufort Sea (fourth row of Fig. 7) where no clouds are observed in both the true-color image (Fig. 7j) and the

classification map (Fig. 7k).

For all four cases, over clear sky pixels, discrimination between open water pixels (blue pixels in Fig. 7b, Fig. 7e, Fig. 7h, and Fig. 7k) and ice pixels (white pixels in Fig. 7b, Fig. 7e, Fig. 7h, and Fig. 7k) based on the $\rho_5$ thresholds coincided well with the locations of open water and ice observed from the true-color images (first column in Fig. 7). Therefore, it can be concluded that the ice-water classification in this study is successfully done and the calculated SICs correspond well to the

classification results (third column in Fig. 7). In addition, cloud pixels only detected from the NDSI criterion (red pixels in second column in Fig. 7) are rarely observable for the cases of C2, C3, and C4 which further demonstrates the validity of the assumption that all cloud pixels had been removed prior to ice/water classification in Section 3.2 for the Landsat-8 scenes under the three categories.

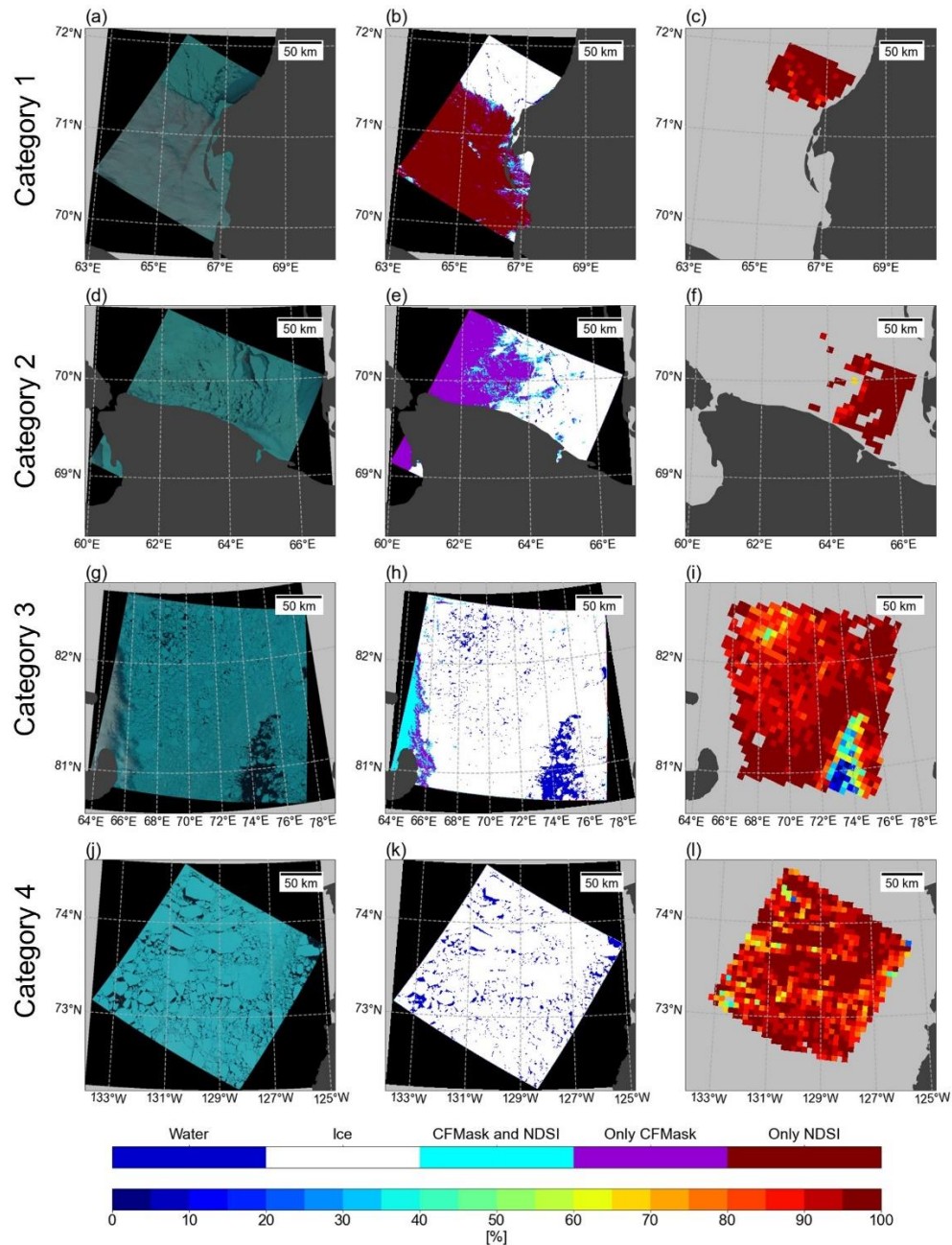

**Figure 7: Example of (a, d, g, i) original Landsat-8 true-color image, (b, e, h, k) classification map of ice (white), open water (blue), and cloud (cyan, purple, and red), and (c, f, i, l) Landsat-8 SICs with 6.25 km resolution on (first row) Mar. 22, 2022 over the Kara Sea, (second row) Mar. 17, 2021 over the Barents Sea, (third row) Jun. 26, 2022 over the Kara Sea, and (fourth row) Jun. 15, 2022 over the Beaufort Sea. From top to bottom row, the select cases correspond to the cloud contamination categories of 1, 2, 3, and 4 respectively. SICs are not estimated over areas of cloud mask (cyan, purple and red pixels in the middle column), and SICs near the coastal area (6.25 km) are masked in this figure. The true-color images were obtained from Earth Resources Observation and Science (EROS) Center (2020). .**


### 4.3 Uncertainty of Landsat-8 SIC

The estimated $\sigma_{\text{SIC}}$ from all the selected 480 scenes in Section 3.4 was less than 1%, and the $\rho_5$ threshold was found to be
responsible for more than 99% of $\sigma_{\text{SIC}}$. The uncertainty (i.e., $\sigma_{\text{SIC}}$) estimated separately for different regions or different cloud contamination categories all remained within 1% and had similar contribution ratios with the $\rho_5$ threshold being the dominant contribution factor to $\sigma_{\text{SIC}}$ (see Table D1, D2, D3 in Appendix D for the exact values). Thus, $\sigma_{\text{SIC}}$ seems to be independent of region or cloud contamination label. However, $\sigma_{\text{SIC}}$ was found to be dependent on the SIC value itself. Figure 8 shows the variation in $\sigma_{\text{SIC}}$ with respect to the SIC range, illustrating that the lowest uncertainty is ~0.2% in the SICs from 0 to 10% and
from 90 to 100% while the highest uncertainty of 4.5% is observed in SIC ranged from 50 to 60% (see Table D4 for exact values). Still, the $\rho_5$ threshold explains most of the uncertainty, regardless of SIC values. In spite of the relatively high uncertainty in Landsat-8 SIC between 20% and 80%, the product can still be used for validation purposes because most PMW SIC products exhibit much larger uncertainties over such SIC range of up to ±12% in the winter (Ivanova et al., 2014) and ±20% in the summer (Meier and Notz, 2010).


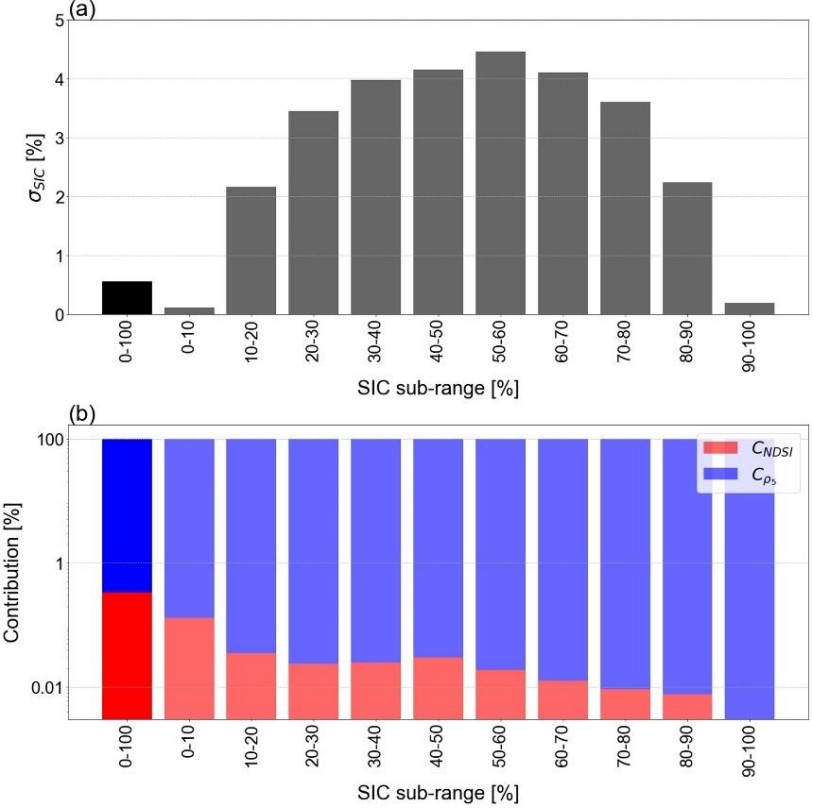

**Figure 8: (a) Uncertainties in Landsat-8 SICs ($\sigma_{\text{SIC}}$) and (b) contributions of the $\rho_5$ (blue) and the NDSI thresholds (red) to the estimated uncertainties for different SIC range, where $\rho_5$ and NDSI are the TOA reflectivity at band 5 of the OLI sensor and the Normalized Difference Snow Index, respectively. Dark and light-coloured bars indicate the uncertainty and contributions**
**computed from all 480 scenes and separately for each SIC sub-range, respectively.**

## 5 Discussion

### 5.1 Possible Errors in SIC Produced from Landsat-8 Images Labelled C1

As mentioned in Sections 3.1 and 3.2, the "clear pixel assumption", which assumes that all cloud pixels in a Landsat-8 image have been removed by the application of CFMask, is not valid for Landsat-8 images labelled C1 in Section 3.1. For Landsat-8 SIC associated with C1 category, therefore, it is necessary to evaluate the possible uncertainty in SIC induced by unremoved cloud pixels (i.e., underestimated cloud cover). Evaluation was performed as follows: From Landsat-8 images under the C1 category, those having 100% cloud cover based on visual inspection, but less than 100% cloud cover from CFMask were selected. From these images, the $\rho_5$ and NDSI values were collected over pixels that were not masked by CFMask (i.e., undetected cloud pixels). Classification following the process illustrated in Fig. 4 was performed over the collected undetected cloud pixels to quantitatively assess the possible errors in SIC estimated over such pixels. A total of 6,721,605 undetected cloud pixels were used in this evaluation, and the name and location of Landsat-8 images used are shown in Fig. S3 and Table S11 in the supplements.

The classification result is summarized in Table 4 and Fig. C2. From the distribution of the unmasked cloud pixels in the two-dimensional histogram between NDSI ($x$-axis) and $\rho_5$ ($y$-axis) in Fig. C2, it can be seen that the NDSI criterion used in this study reinforces the cloud removal process by filtering cloud pixels that were undetected by CFMask. However, even with the additional procedure to remove remained cloud signals (i.e., the NDSI criterion), 8.54% of the undetected cloud pixels are still classified as open water and/or ice. It is noted that such cloud pixels (i.e., cloud pixels undetected from both CFMask and the NDSI criterion) were predominantly classified as ice (Table 4). Therefore, it can be inferred that the undetected cloud pixels in a Landsat-8 image can induce positively biased SIC and thus for SICs produced from Landsat-8 images labelled C1 over which the "clear pixel assumption" is invalid, the error from ice/water classification is estimated to be large as 8.54% from the percentage of cloud pixels classified as ice in Table 4. The possibility of such large uncertainties should be taken into note when using Landsat-8 SIC labelled C1.

| Classified ($\rho_5$ and NDSI) Reference | Open water | Ice | Cloud |
|---|---|---|---|
| Undetected Cloud (by CFMask) | 215 (0.00%) | 573,922 (8.54%) | 6,147,468 (91.46%) |

**Table 4: The number of cloud pixels that were undetected from** the C Function of Mask (**CFMask**) classified into open water, ice, and cloud from application of the procedure in Fig. 4. The scenes used for the evaluation belong to C1 (i.e. underestimated cloud cover) category from the method described in Section 3.2. The values inside the parentheses indicates the percentage of pixels that belong to each category.

## 5.2 Evaluation of Landsat-8 SIC Using Ice Charts

The accuracy of Landsat-8 SIC produced in this study was evaluated using ice charts provided by MET Norway as reference. For quantitative comparison, ice charts were collocated into the grid system of Landsat-8 SIC (i.e., PSR grid with 6.25 km resolution) as follows: Data points on the ice chart within each 6.25 km × 6.25 km grid cell were collected, and the SIC mean value from the collected data points were taken as the representative SIC value of ice chart for the 6.25 km × 6.25 km grid cell. It is important to be noted that SIC values in the original ice charts are not normally-defined SICs in satellite remote sensing, but contain uncertainties represented by the ice concentration range defined in Table A1. Therefore, it is necessary to consider the propagation of uncertainty in the collocation process. Uncertainty of the collocated ice chart SIC was estimated by taking the average of the uncertainty in ice chart data points collected from each 6.25 km × 6.25 km grid cell. To avoid the influence of land contamination, a 6.25 km coastal area mask was applied to both SICs prior to the comparison. The number of collocated data points used in the evaluation was 45,547.

From Fig. 9, a good linear relationship (i.e., correlation coefficient of 0.96) between Landsat-8 SIC and ice chart SIC is observed. The spread (i.e., 20 and 80 percentiles) of Landsat-8 SIC for ice chart SIC sub-ranges, which are shown as red vertical lines in Fig. 9a, was larger in SIC ranged from 20% to 80% relative to other ranges, which is likely due to a consequence of the wider concentration intervals assigned to the 20-80% of SIC values in the original ice chart (Table A1). In addition, SIC from the ice charts tends to be higher than that found from Landsat-8 SIC and the bias is more pronounced in the lower SIC range. This type of state dependent overestimation of SIC from ice charts has been reported in previous works of Tonboe et al. (2016) and Cheng et al. (2020), which shows that overestimation of SIC from ice charts is largest in the lower SIC range due to the "better-safe-than-sorry" practices of the ice charting community. For quantitative comparison of the bin-wise mean biases in Landsat-8 SIC relative to ice chart SIC, bin-averaged SICs from Landsat-8 (red triangle in Fig. 9b) and from ice charts (blue circles in Fig. 9b) were plotted along with their respective uncertainties. Uncertainties of Landsat-8 SICs over the SIC sub-ranges were taken as the values from Table D4. Except for 70-80% SIC interval, Landsat-8 SICs were negatively biased to ice chart SICs. However, the mean biases for all SIC sub-range were found to be within the uncertainty ranges estimated for each product.

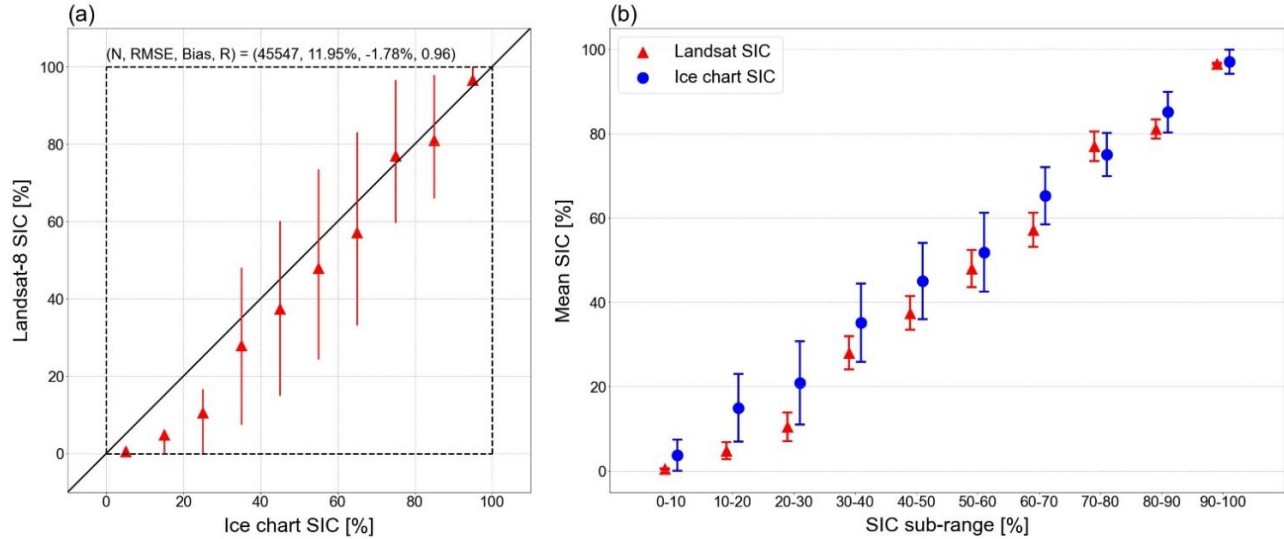


**Figure 9: (a)** Scatter plot of bin-wise mean Landsat-8 SICs and ice chart SIC sub-ranges. The bin-wise mean SICs are shown as red triangles, and the 20 and 80 percentiles are shown as the red vertical lines. The values for number of data points (N), root-mean-square error (RMSE), bias, and Pearson correlation coefficient (R) are presented. **(b)** For the same SIC intervals as (a), the bin-wise mean SICs of Landsat-8 (red triangle) and ice chart (blue circle) and their respective uncertainties (vertical lines). The
uncertainties of Landsat-8 SIC are taken from the values in Table D4.

## 5.3 Evaluation of Landsat-8 SIC Over Melt Ponds

The evaluation of variation in Landsat-8 SICs due to melt pond presence was performed using the MPF dataset (Niehaus and Spreen, 2022) described in Section 2.5 as reference data for melt ponds. As mentioned in Section 2.5, a total of six Landsat-8
scenes obtained from the periods of Jul. 2020, Aug. 2020, and Jul. 2021 were used in the evaluation. The evaluation was conducted as follows: First, the collocation of the MPF dataset into the grid system of Landsat-8 SIC was performed. This was done by collecting MPF data points within each 6.25 km × 6.25 km grid cell and taking the mean value of the collected MPF data points as the MPF values for each corresponding grid cell in 6.25 km resolution. In addition, from the OW masks in the MPF dataset, SIC values ($SIC_{MPF}$) were computed in the grid system of Landsat-8 SIC following the same method
introduced in Section 3.3. Second, in order to remove the effects of SIC variation from the evaluation, the corresponding Landsat-8 SIC and MPF data points were collected when data points satisfy $SIC_{MPF}$=100%. The number of collected data points is 98. From the collected data points, the net ice surface fraction ($C_{net}$) was computed as the following,

$$C_{net} = (1 - MPF) \times SIC_{MPF} \tag{10}$$

where MPF is the melt pond fraction and $SIC_{MPF}$ is the estimated SIC value from the MPF dataset. Since $SIC_{MPF}$ was fixed to
100%, in this study, the variation in $C_{net}$ can be considered solely driven by the variation in MPF.

The robustness of the Landsat-8 SIC to the presence of melt ponds is illustrated in Fig. 10, which is a scatter plot between the collected $C_{net}$ (x-axis) and Landsat-8 SIC (y-axis). In this plot, the MPF was varying from 0% to 33%, and therefore, the

computed values of C_net are ranged from 67% to 100%. However, SICs estimated from Landsat-8 are observed to be nearly independent to the varying C_net (statistically insignificant correlation coefficient of 0.11) and thus nearly independent of MPF.

Although a few Landsat-8 SICs are observed to be affected by melt pond presence (data points highlighted in red from Fig. 10), which can be expected because melt ponds are not easily distinguished from open water, the number of such data points are very small (only four data points out of 98 data points). It is noted that on average the deviation from 100% ice concentration computed from the data points shown in Fig. 10 was less than 1%. Therefore, it can be inferred that the impact of melt pond presence in SIC calculation using Landsat-8 imagery is small, and that the proposed algorithm for SIC

production in this study is robust regardless of surface melting.

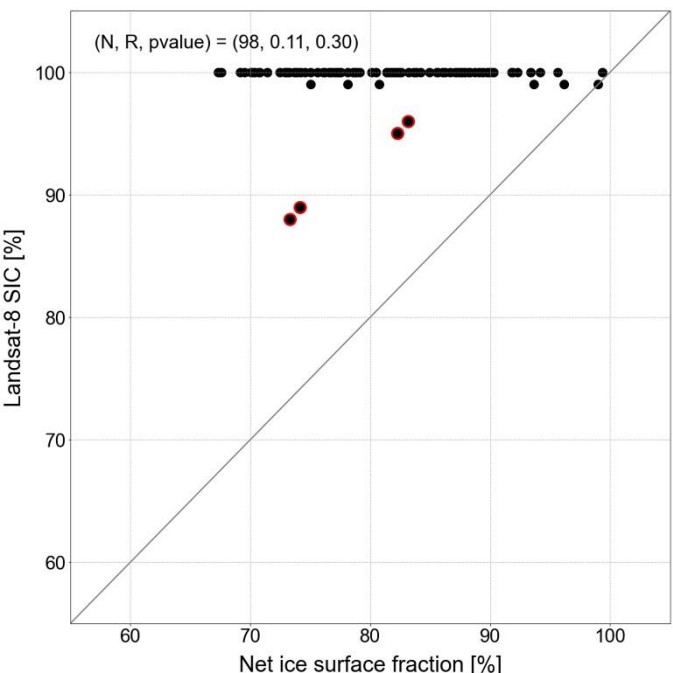

**Figure 10: Scatter plot of net ice surface fraction (x-axis) and Landsat-8 SIC (y-axis). The data points shown satisfy SIC_MPF=100%**
**and have MPF that vary from 0% to 33%. Data points with more than 4% deviation of Landsat-8 SIC from 100% ice concentration are highlighted in red. The values for number of data points (N), Pearson correlation coefficient (R), and p-value for the correlation coefficient are presented.**

## 5.4 Possible Applications of Landsat-8 SIC for Assessing PMW-based SICs

Landsat-8 SIC produced from this study can be utilized to assess the PMW-based SICs. This section provides the possible application of the constructed Landsat-8 SIC for examining PMW-based SICs. To do this, Landsat-8 SIC was downscaled to 25 km resolution and compared against SICs estimated from BT and NT algorithms, both provided in the PSR grid with 25

km resolution and obtained from NSIDC (Meier et al., 2021), for the selected two cases of Landsat-8 scenes acquired during melting (Jul. 21, 2022 over the Laptev Sea) and freezing (Mar. 4, 2022 over the Chukchi Sea) seasons, respectively. To

avoid the influence of land contamination, a coastal area mask, which was also downscaled to 25 km resolution, was applied before the comparison.

Figure 11 illustrates spatial distributions of the three different SICs, differences in SICs of BT and NT from Landsat-8, and scatter plots of BT and NT SIC against Landsat-8. For the case of the melting season (top two rows in Fig. 11), BT SIC showed a positive bias of 8.95%, RMSE of 16.30%, and correlation coefficient of 0.92 (Fig. 11d) to Landsat-8 SIC while

SIC retrieved from the NT algorithm is negatively biased to Landsat-8 SIC by -5.21% with a RMSE of 14.35%, and correlation coefficient of 0.94 (Fig. 11h). It is interesting to note that BT SICs are positively (negatively) biased to Landsat-8 SIC for lower (higher) concentrated ice areas (Fig. 11c), while opposite patterns are observed for NT SICs (Fig 11g). Both PMW-based SICs show the largest disagreement with Landsat-8 SIC near the edges of pack ice (i.e., boundaries between sea ice and open water).

For the scene in the freezing season (bottom two rows in Fig. 11), the BT and NT algorithms produced nearly 100% SICs for all grids in this case while Landsat-8 SIC shows lower SIC values in regions coinciding with the leads in the pack ice observed from the true-color image. As a result, positive biases were observed near the position of the lead (Fig. 11k and Fig. 11o), and mean biases for the BT and NT algorithms were 0.83% and 0.53%, respectively. RMSEs of BT and NT SIC were calculated as 1.35% and 0.81%, respectively, which are lower than the RMSE evaluated during the melting season for the

two SIC algorithms (Fig. 11i and Fig. 11p).

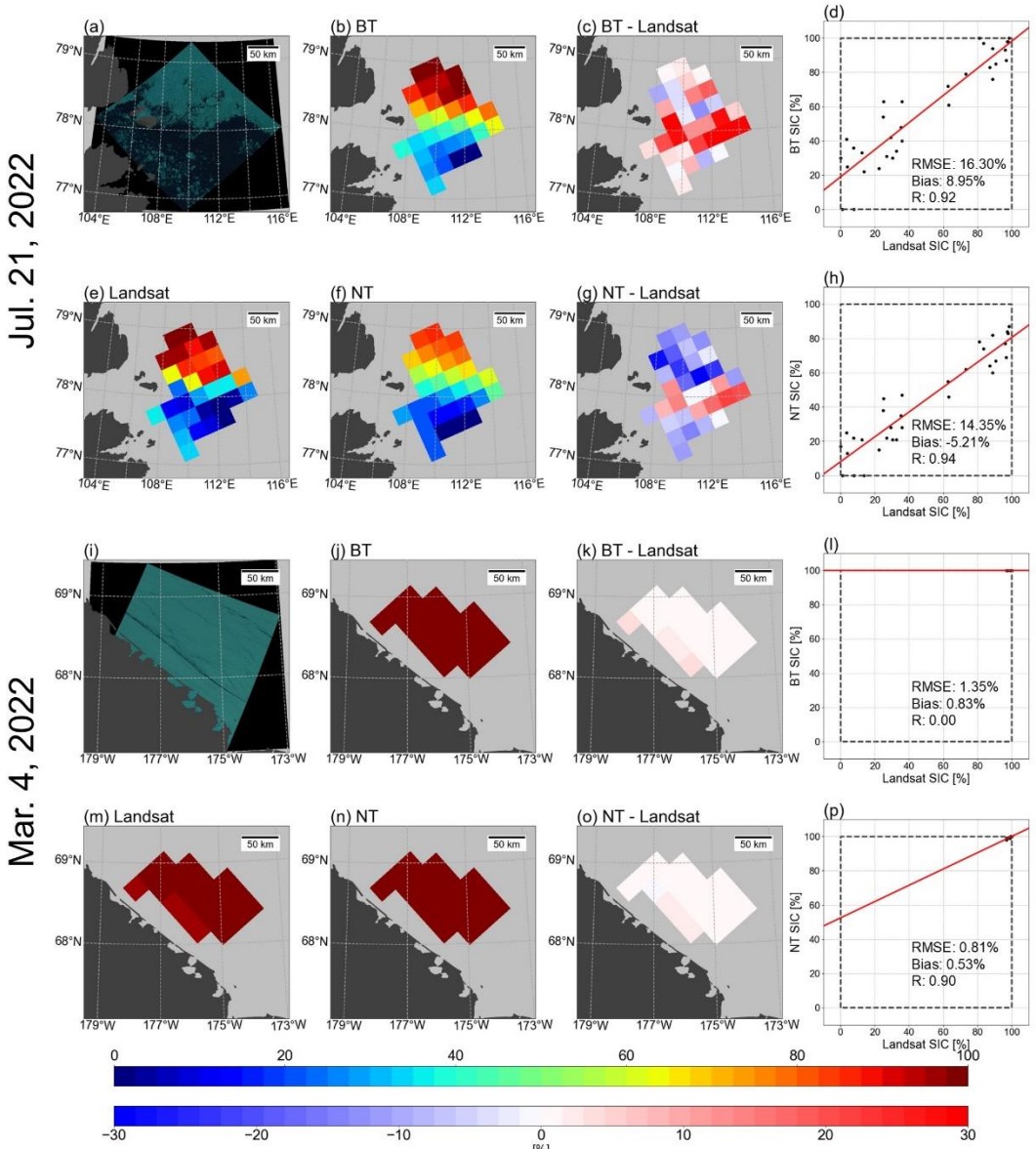

**Figure 11: Geographical distributions of (a, i) original Landsat-8 true-color image, (e, m) Landsat-8 SIC, (b, j) SIC from the Bootstrap (BT) algorithm, (f, n) SIC from the NASA Team (NT) algorithm, (c, k) difference in SICs between BT and Landsat-8, (g, o) difference in SICs between NT and Landsat-8 and scatterplot (d, l) between Landsat-8 SIC and SIC from BT and (h, p) between Landsat-8 SIC and SIC from NT. The values of root-mean-square error (RMSE), bias, and Pearson correlation coefficient (R) are presented with the scatter plots. Upper two panels for July 21, 2022 (melting season) over the Laptev Sea and for March 4, 2020 over the Chukchi Sea, respectively. The true-color images were obtained from Earth Resources Observation and Science (EROS) Center (2020) and the SIC retrievals from the BT and NT algorithms were obtained from Meier et al. (2021).**

**6 Code and Data Availability**

The Landsat-8 SIC dataset can be downloaded at https://zenodo.org/doi/10.5281/zenodo.10973297 (Jung et al., 2024). Datasets generated for each Arctic region can be found in "sic_landsat08_{region name}.nc. The datasets are stored in netCDF format and can be accessed using software including Python, MATLAB, and QGIS. Along with the SIC values, N, coastal mask, and region mask are also provided in a $1792 \times 1216$ array format. Cloud contamination category and name of the original Landsat-8 files are also provided for each scene. Variables in the netCDF file are visualized in Fig. 12. Fill values were assigned to grids outside the coverage of a Landsat-8 scene, grids over land, or grids masked by clouds (black grids in Fig. 12a, b). Description of each variable and the fill/flag values are summarized in Table 5.

Datasets used to produce and validate the Landsat-8 SIC are listed as follows.

- 'Landsat-8 Collection 2 Level 1 Product' and the corresponding true-color images are accessible from United States Geological Survey Earth Explorer at https://earthexplorer.usgs.gov/ (Earth Resources Observation and Science (EROS) Center, 2020).

- 'Arctic and Antarctic Regional Masks for Sea Ice and Related Data Products, Version 1' used to mask non-ocean areas and distinguish regions can be accessed at https://doi.org/10.5067/CYW3O8ZUNIWC (Meier and Stewart, 2024).

- 'Landsat surface type over water from supervised classification of surface broadband albedo estimates (Version_2021_fv0.01)' used to test the performance of the ice and open water classification can be accessed at http://doi.org/10.25592/uhhfdm.9181 (Kern, 2021).

- 'Arctic Ocean – Sea Ice Concentration Charts – Svalbard and Greenland' ice charts used to evaluate the produced Landsat-8 SIC can be accessed at https://doi.org/10.48670/moi-00128 (E.U. Copernicus Marine Service Information (CMEMS), accessed on 11 June 2024).

- 'Melt pond fraction on Arctic sea-ice from Sentinel-2 satellite optical imagery (2017-2021)' used to test the robustness of Landsat-8 SIC over melt ponds can be accessed at https://doi.org/10.1594/PANGAEA.950885 (Niehaus and Spreen, 2022).

- 'NOAA/NSIDC Climate Data Record of Passive Microwave Sea Ice Concentration, Version 4' used to illustrate possible applications of Landsat-8 SIC dataset can be accessed at https://doi.org/10.7265/efmz-2t65 (Meier et al., 2024).

The python codes for Landsat-8 SIC production, sensitivity and uncertainty analysis, ice/water classification evaluation, Landsat-8 SIC validation, and figure generation are accessible at https://doi.org/10.5281/zenodo.12754603 (Jung, 2024). Example data to check the functionalities of each python code are provided with the code repository.

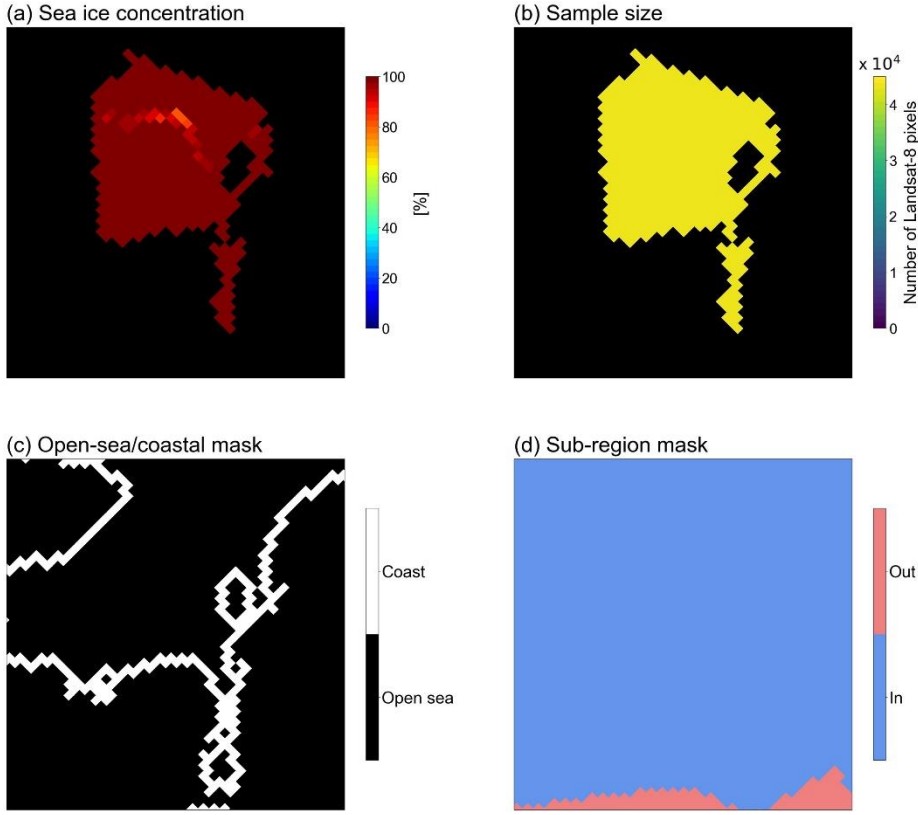


**Figure 12: Variables in the Landsat-8 SIC netCDF. The scene is from Jun. 12, 2021 over the Canadian Archipelago. For the (d) sub-region mask, 'In' and 'Out' denote grid cells located inside and outside the designated region, respectively.**


| Variable | Long Name | Flag values |
|---|---|---|
| sea_ice_concentration | Estimated fractional sea ice area from Landsat-8 measurements | [-99: Fill value] |
| sample_size | Number of Landsat-8 pixels used to estimate the sea ice concentration | [0: Fill value] |
| coastal_mask | Open sea/Coastal Flag | [0: Open_sea, 1: Coast] |
| sub_region_mask | Sub-region Flag | [0: inside_sub_region, 1: outside_sub_region] |

**Table 5: Variables in the Landsat-8 SIC netCDF file, name of the variables, and the flag values for each variable.**

**7 Conclusion**

In this study, three years (2020-2022) of Landsat-8 data were collected and used to generate sea ice concentration (SIC)
datasets in the polar stereographic grid with 6.25 km resolution. A total of 14,297 Landsat-8 images were used to calculate
2,934,399 SIC grid points. Each Landsat-8 SIC is catalogued under a netCDF file named after the twelve regions.

Each Landsat-8 image was labelled into four cloud contamination categories (i.e., C1, C2, C3, and C4) according to the
overall quality of cloud mask over the image. The categories are provided in the variable under the name
'cloud_contamination_category' of the Landsat-8 SIC dataset to allow for selection of SICs calculated without the
interference of cloud signals.

The uncertainty of Landsat-8 SIC was estimated to be ranged from 1 to 4% based on the Gaussian error propagation method.
In addition, to regulate the potential uncertainty that may arise from the use of partially-covered grid cells, SIC was only
produced for grid cells with over 99% of its area covered by Landsat-8 pixels. Evaluation of Landsat-8 SIC using SIC from
ice charts show good linear correlation between the two products and also reveal existence of negative bias in Landsat-8 SIC.
However, the bias was found to be within the uncertainty range of the Landsat-8 and ice chart SIC. In addition, the
production method used for Landsat-8 SIC was found to be robust over melt ponds. Thus, Landsat-8 SIC produced in this
study can be considered to be reliable estimates of SIC.

Comparison of Landsat-8 SIC against SIC retrievals from NASA Team (NT) and Bootstrap (BT) algorithms for two cases
reveal overall negative bias in NT and positive bias in BT SIC. The spatial distribution of the bias shows that bias in NT and
BT SIC may be related to the SIC values, with NT SIC exhibiting stronger negative bias in high SIC regime, and BT SIC
showing stronger positive bias in low SIC regime. This suggests that the Landsat-8 SIC can be used as reference SIC to
generate quantitative error statistics of various passive microwave SIC retrievals over different regions, seasons, and SIC
values, which can be used to develop an optimal combination of existing SIC algorithms or be used to provide realistic
observation errors to enhance the performance of sea ice data assimilation.

Future works are aimed to extend the temporal and spatial coverage of the current Landsat-8 SIC dataset by the addition of
Landsat-8 images from the years 2018 and 2019. In addition, given the large number of Landsat-8 SIC data points generated
in this study, the obtained SIC values also have the potential to be used to train deep-learning models in order to retrieve
optimal SIC estimates over the Arctic.



# 8 Appendices

## Appendix A: SIC Values in The MET Norway Ice Chart

| Concentration class | Concentration interval [%] | Fixed concentration value [%] | Concentration range [%] |
|---|---|---|---|
| Fast Ice | 100 | 100 | 0 |
| Very Close Drift Ice | 90-100 | 95 | 5 |
| Close Drift Ice | 70-80 | 75 | 5 |
| Open Drift Ice | 40-60 | 50 | 10 |
| Very Open Drift Ice | 10-30 | 20 | 10 |
| Open Water | 0-10 | 5 | 5 |

**Table A1: Concentration class, concentration interval, fixed concentration value, and concentration range of the operational ice chart produced by MET Norway.**


## Appendix B: Visual Inspection for Cloud Mask Quality Control

In this section, a step-by-step description of the process taken to perform the visual inspection of Landsat-8 scenes is presented. As defined in Section 3.1, each pixel in a Landsat-8 scene can be sorted into the following four categories depending on the state of cloud mask for the pixel: False negative (FN; cloud pixel mistaken as clear pixel), false positive (FP; clear pixel mistaken as cloud pixel), true negative (TN; clear pixel identified as clear pixel), and true positive (TP; cloud pixel identified as cloud pixel). It is noted that the pixels with FN are used to calculate SICs while the pixels with FP are not, indicating that the presence of FN pixels can directly introduce errors in the calculated SIC value. Therefore, visual inspection was performed very strictly to detect FN pixels.

Figure B1 outlines the steps taken to perform the visual inspection. The descriptions of each step are provided along with an example case of a Landsat-8 scene that is categorized into C1 during the section-wise inspection stage.

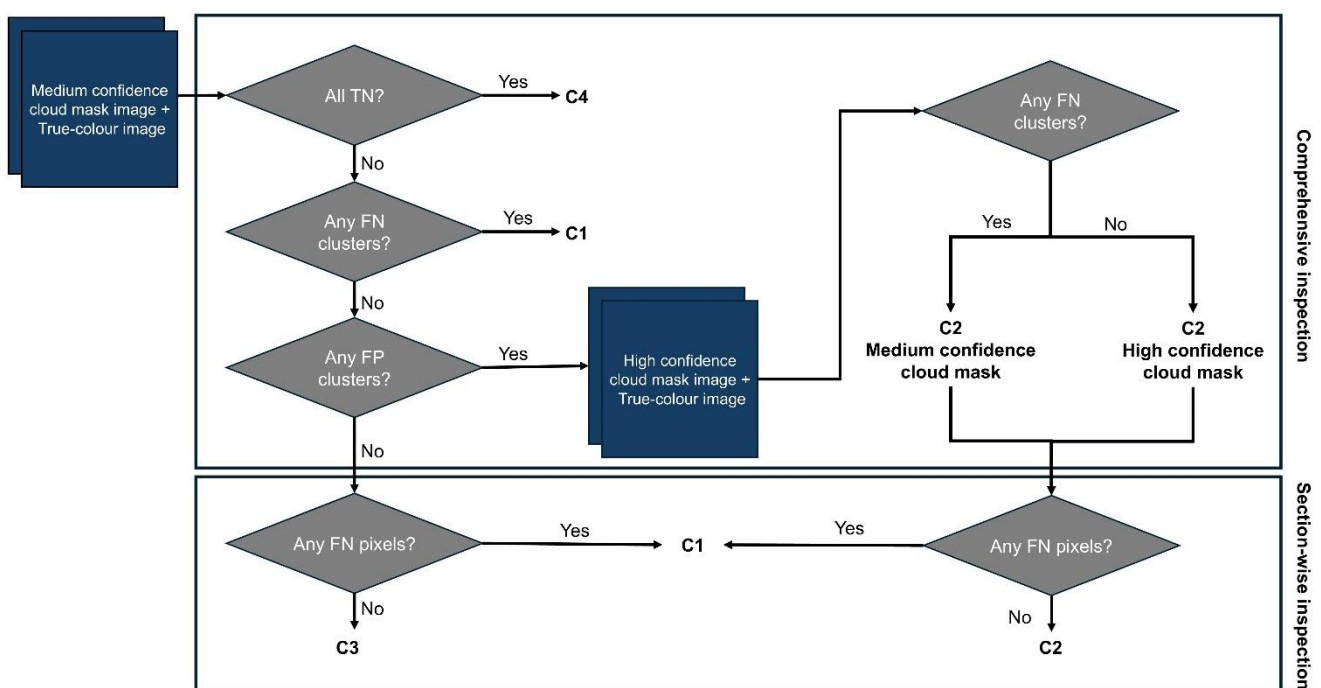

**Figure B1: Processing pipeline of the visual inspection step. Each Landsat-8 image is labelled as C1 (i.e., underestimated cloud cover), C2 (i.e., overestimated cloud cover), C3 (i.e., correctly estimated cloud cover for cloudy sky), or C4 (i.e., correctly estimated cloud cover for clear sky) depending on the observed dominance of true negative (TN; clear pixels identified as clear pixels), false** 700 **negative (FN; cloud pixels mistaken as clear pixels), false positive (FP; clear pixels mistaken as cloud pixels), and true positive (TP; cloud pixels identified as cloud pixels) pixels.**

Step 1. Generating jpeg file of cloud mask (i.e., cloud mask image).

For each Landsat-8 scene, a false-colour image with each pixel classified as ice (white pixels in Fig. B2b, d), open
water (blue pixels in Fig. B2b, d), cloud (grey pixels in Fig. B2b, d), and fill value (black pixels in Fig. B2b, d) is
constructed using the OpenCV module in Python. Ice and open water pixels are differentiated using the method
described in Section 3.2. Cloud pixels are classified by masking the medium confidence cloud, high confidence
cirrus, cloud shadow, and dilated cloud pixels identified by the quality assessment band (i.e., the cloud mask array
produced by CFMask).

Step 2. Comprehensive inspection of cloud mask quality.

The cloud mask image generated in Step 1 is visually inspected against the true-colour image to identify sections
populated with FN, FP, TN, or TP pixels. This is done in the following order: First, if no cloud pixels are observed
from both the cloud mask image and the true-colour image (i.e., all pixels in the image are TN pixels), the scene is
labelled as C4. Second, if any cluster of FN pixels is observed, the scene is labelled C1. Third, if any cluster of FP
pixels is identified, the scene is labelled C2 and passed on to Step 3. If the clusters of cloud pixels in the cloud mask

image are well corresponding to the position of clouds observed in the true-colour image (i.e., TP pixels), the scene is labelled C3 and passed on to Step 4 (Fig B2a, b).

Step 3. Comprehensive inspection of cloud mask quality for C2.

For the scenes passed on to this step (i.e., scenes labelled C2 from Step 2), the cloud mask image is recreated using a higher confidence threshold (i.e., high confidence cloud, high confidence cirrus, cloud shadow, and dilated cloud pixels) for the quality assessment band. The new cloud mask image is visually inspected against the true-colour image, and if any cluster of FN pixels is observed, the confidence threshold for the quality assessment band is returned to its initial value (i.e., medium confidence cloud, high confidence cirrus, cloud shadow, and dilated cloud pixels). If the observed clusters of cloud pixels in the new cloud mask image are well corresponding to the position of clouds observed in the true-colour image, the higher confidence threshold is kept, and the scene is passed on to Step 4.

Step 4. Section-wise inspection of cloud mask quality.

In this step, the identified clusters of TP pixels are inspected in more detail. For each cluster of TP pixels observed, we zoom in (i.e. about $1000 \times 1000$ pixels; the full-size image is approximately $8000 \times 8000$ pixels) to the section of the cluster to check for the existence of FN pixels. If any FN pixels are found within the cluster, the scene is labelled C1 (Fig B2c, d).

An example of how a Landsat-8 scene may be categorized according to the process described in Fig B1 is presented using the case of a Landsat-8 scene acquired on Mar. 25, 2022, over the Barents Sea (Fig B2). First, from Step 2. (i.e., the comprehensive inspection step), visual inspection of the cloud mask image (Fig. B2b) against the true-colour image (Fig. B2a) shows that the position of clouds in the cloud mask array is generally well corresponding to those observed in the true-colour image. Therefore, at this step, this scene is labelled C3 and passed on to Step 4 as described above. Next, the section-wise inspection of the cloud mask quality is performed by zooming in to the cloud areas. This is illustrated in Fig. B2c and Fig. B2d, which is a zoomed in image of the area enclosed by the red rectangle in Fig. B2a and Fig. B2b. Inspection of this sub-section shows the presence of unmasked cloud shadow pixels, which results in the erroneous classification of ice as open water. Therefore, at this step, the label of this scene is changed to C1.

The visual inspection was done by HJ and it took approximately 5 – 10 minutes to inspect one Landsat-8 scene for cloud cover.

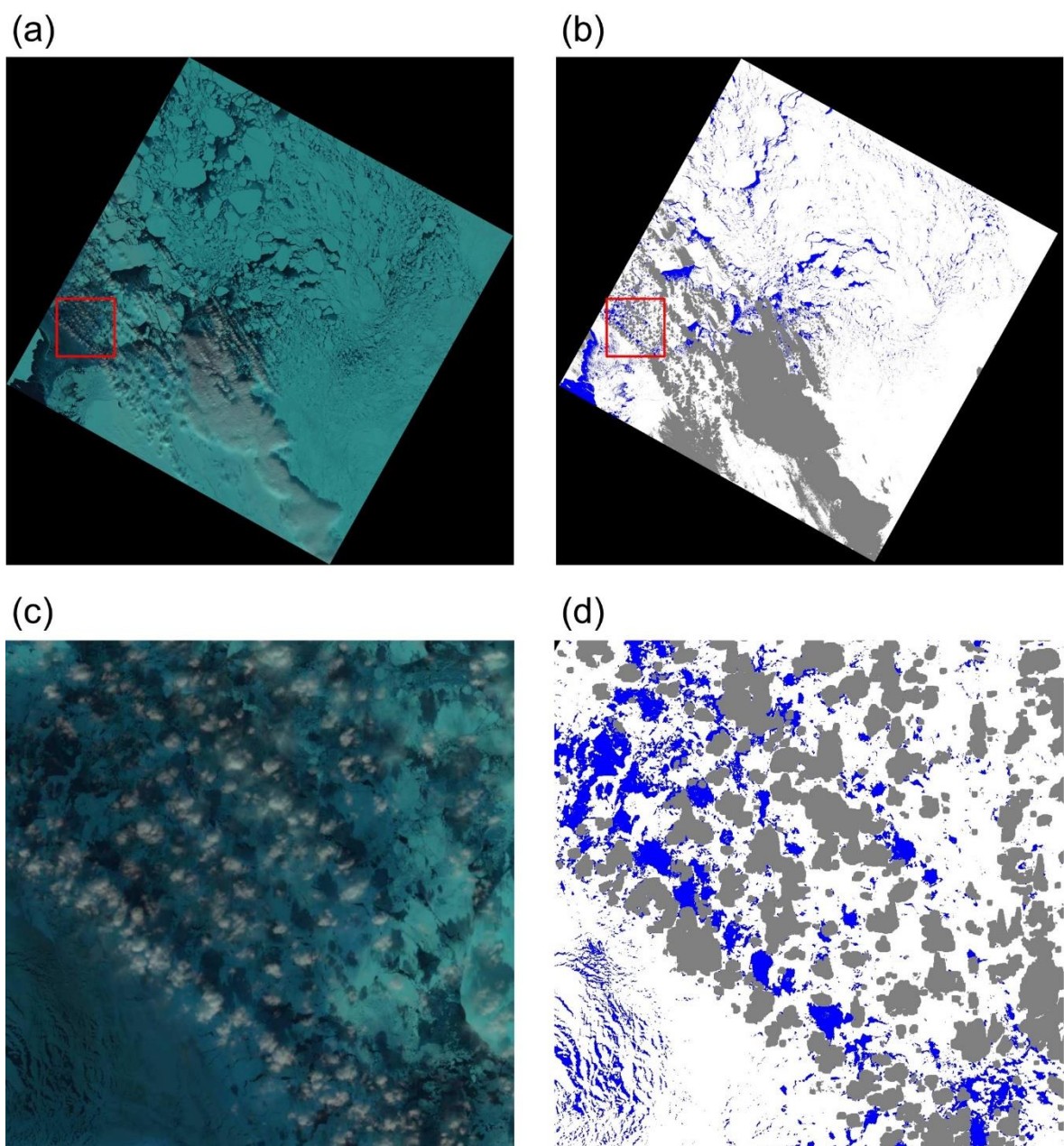

**Figure B2: The case of a Landsat-8 scene classified as C1 (i.e., underestimated cloud cover) from the visual inspection step. Shown in the panels are (a) full size true-colour image, (b) full size cloud mask array, (c) true-colour image of the area enclosed by the red rectangle in (a) and (b), and (d) cloud mask array of the area enclosed by the red rectangle in (a) and (b). The blue, white, grey, and black pixels in (b) and (d) are open water, ice, cloud, and fill value pixels, respectively. The scene is from Mar. 25, 2022, over the Barents Sea. The true-colour image is obtained from Earth Resources Observation and Science (EROS) Center (2020).**


 **Appendix C: Validation of Ice and Open Water Classification**

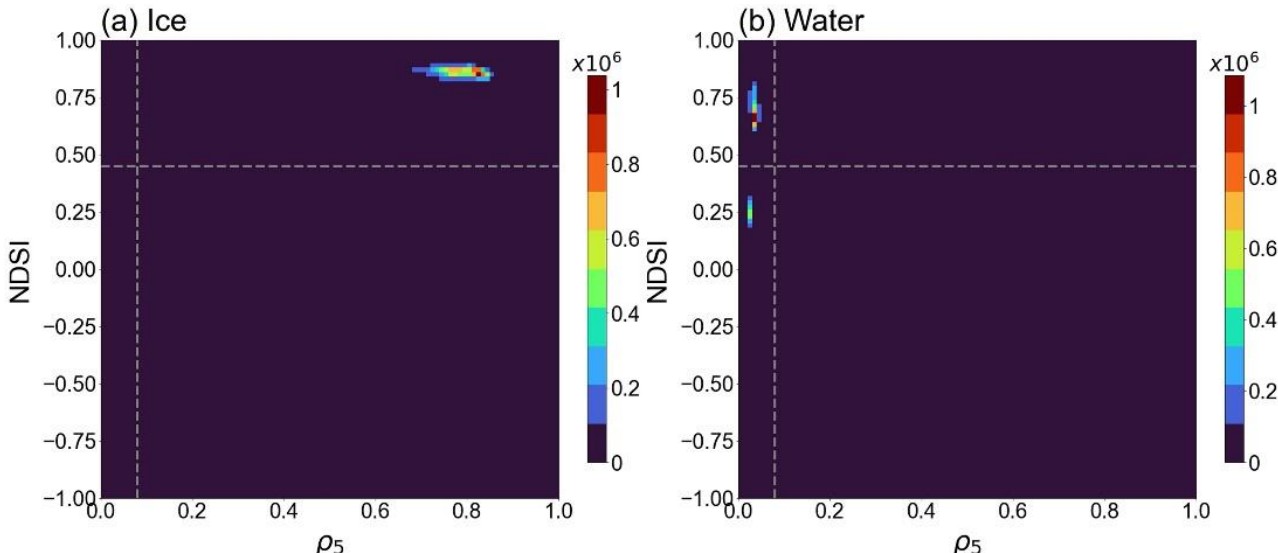

**Figure C1: Scatterplot between NDSI and $\rho_5$ for (a) ice and (b) open water. The values for ice and open water pixels were collected using the ice/water surface classification map (Kern, 2021) as reference data. The thresholds for NDSI and $\rho_5$ used in this study are shown by the white dashed lines. The colorbars denote the number of pixels.**


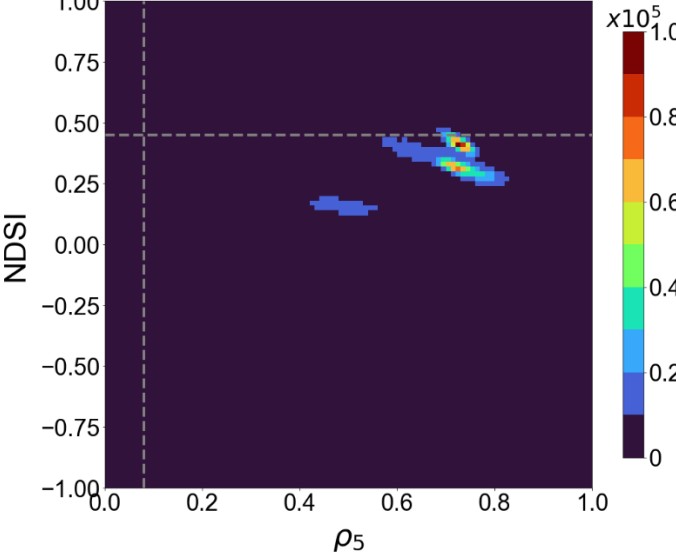

**Figure C2: Scatterplot between NDSI and $\rho_5$ for cloud pixels that remain unmasked after the application of CFMask. The pixels are acquired from ten select Landsat-8 images categorized as C1. The thresholds for NDSI and $\rho_5$ used in this study are shown by the white dashed lines. The colorbars denote the number of pixels.**


## Appendix D: Uncertainty of Landsat-8 SIC With Respect To Region, Cloud Contamination Label, an SIC Sub-Range

| $C_{\rho5}$ [%] | $C_{NDSI}$ [%] | $\sigma_{SIC}$ [%] |
|---|---|---|
| 99.66 | 0.34 | 0.56 |

**Table D1: Contributions from $\rho_5$ ($C_{\rho5}$) and NDSI ($C_{NDSI}$) to the estimated uncertainty of SIC ($\sigma_{SIC}$) and $\sigma_{SIC}$ over all 480 scenes.**


| Region | $C_{\rho5}$ [%] | $C_{NDSI}$ [%] | $\sigma_{SIC}$ [%] |
|---|---|---|---|
| Baffin Bay | 99.74 | 0.26 | 0.80 |
| Barents Sea | 99.84 | 0.16 | 0.65 |
| Beaufort Sea | 99.33 | 0.67 | 0.30 |
| Bering Sea | 97.46 | 2.54 | 0.89 |
| Canadian A. | 99.93 | 0.07 | 0.25 |
| Central Arctic | 99.86 | 0.14 | 0.56 |
| Chukchi Sea | 99.97 | 0.03 | 0.45 |
| E. Greenland | 99.58 | 0.42 | 0.65 |
| E. Siberian | 100.00 | 0.00 | 0.53 |
| Hudson Bay | 98.48 | 1.52 | 0.54 |
| Kara Sea | 99.23 | 0.77 | 0.43 |
| Laptev Sea | 99.99 | 0.01 | 0.63 |

**Table D2: Contributions from $\rho_5$ ($C_{\rho5}$) and NDSI ($C_{NDSI}$) to the estimated uncertainty of SIC ($\sigma_{SIC}$) and $\sigma_{SIC}$ for the twelve regions.**

| Cloud Contamination Category | $C_{\rho5}$[%] | $C_{NDSI}$ [%] | $\sigma_{SIC}$ [%] |
|---|---|---|---|
| C1 | 97.10 | 2.90 | 0.65 |
| C2 | 100.00 | 0.00 | 0.50 |
| C3 | 99.99 | 0.01 | 0.66 |
| C4 | 100.00 | 0.00 | 0.40 |

**Table D3: Contributions from $\rho_5$ ($C_{\rho5}$) and NDSI ($C_{NDSI}$) to the estimated uncertainty of SIC ($\sigma_{SIC}$) and $\sigma_{SIC}$ for the three cloud**
**contamination categories.**

| SIC sub-range [%] | $C_{\rho 5}$ [%] | $C_{NDSI}$ [%] | $\sigma_{SIC}$ [%] |
|---|---|---|---|
| 0-10 | 99.87 | 0.13 | 0.11 |
| 10-20 | 99.96 | 0.04 | 2.16 |
| 20-30 | 99.98 | 0.02 | 3.45 |
| 30-40 | 99.97 | 0.03 | 3.99 |
| 40-50 | 99.97 | 0.03 | 4.15 |
| 50-60 | 99.98 | 0.02 | 4.46 |
| 60-70 | 99.99 | 0.01 | 4.11 |
| 70-80 | 99.99 | 0.01 | 3.61 |
| 80-90 | 99.99 | 0.01 | 2.25 |
| 90-100 | 100.00 | 0.00 | 0.19 |

**Table D4: Contributions from $\rho_5$ ($C_{\rho 5}$) and NDSI ($C_{NDSI}$) to the estimated uncertainty of SIC ($\sigma_{SIC}$) and $\sigma_{SIC}$ for varying SIC sub-range.**

**Author Contribution**

HJ contributed to the conceptualization, investigation, methodology, and software of the research, and wrote the original manuscript. SL contributed to the conceptualization and methodology of the research, review & editing of the manuscript, and provided supervision throughout the research progress. JK contributed to the funding acquisition and review & editing of the manuscript. KL contributed to the conceptualization and review & editing of the manuscript.

**Competing Interests**

The authors declare that they have no conflict of interest.

**Acknowledgements**

This work was supported by Korea Polar Research Institute (KOPRI) Grant entitled "Development and Application of the Earth System Model-Based Korea Polar Prediction System (KPOP-Earth) for the Arctic and Midlatitude High-Impact Weather Event" funded by the Ministry of Oceans and Fisheries under Grant KOPRI PE23010.

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
