# Peer review of "An Arctic sea ice concentration data record on a 6.25 km polar stereographic grid from three-years' Landsat-8 imagery"

_Earth System Science Data, 2024_

## Referee Comment (RC1)

essd-2024-264.    Manuscript review

Title: An Arctic sea ice concentration data record on a 6.25 km polar stereographic grid from three-years' Landsat-8 imagery
Author(s): Hee-Sung Jung et al.
MS No.: essd-2024-264
MS type: Data description paper

**General comments**

The article presents a data set of high resolution and high quality sea ice concentration maps derived from Landsat data that can be used to validate algorithms for ice concentration fields derived from other satellite data.  The authors note that the data should find utility as a benchmark data set by which to judge the accuracy of passive microwave-derived ice concentration data. While the spatial extent of each classified Landsat image is quite small relative to a passive microwave ice concentration field, the large number of classified Landsat images should somewhat make up for this limitation.

The data and methods are new, while having been built upon established methods for classifying visible and near-IR data.  Methods are described in detail with sufficient references. Error estimates and sources of error are given and discussed in the article.   The data are accessible and can be downloaded from the linked Zenodo site. There is one NetCDF file for each of 12 regions.  I plotted SIC fields from the Beaufort region file using Panoply, and displayed a number of image days at random. These looked reasonable, but I did not assess the data quality beyond that. I'll note that the summary on the Zenado site needs significant copy editing.

I rate the data set as excellent in terms of its uniqueness, usefulness, and completeness.

The article is well-structured, and overall, the presentation quality is very good, but in some places it lacks clarity.   I have made comments under Technical Corrections that may help the authors improve clarity.

**Specific comments**

Section 3.1 says that an addition step in cloud mask quality assessment was to visually compare the cloud mask array from each image with the corresponding true-color image.  Please provide more information on this process, including who did the visual inspection and approximately how much time it took for each image. More that 15,000 images are a lot to visually screen, and this step should be described in more detail.

**Technical corrections or wording suggestions**

*Line      Comment or suggested rewording*

36      Remove "at least"

54      There have been developed various PMW SIC algorithms >> Various PMW SIC algorithms have been developed

72      However, there exist discrepancies >> However, discrepancies exist

78      What is meant by "ice/water mixtures"? Note that the differences in algorithms isn't due to the presence of these things, but rather due to the differing sensitivity of the algorithms to these things, so it may be helpful to reword this sentence to reflect that.

88      It would be helpful add a sentence here that notes that you will use the Kern et al. data to validate your own data as described in Section 2.3.

107     The use of "sub-region" confused me. Here, and for most if not all other occurrences throughout the paper, "region" would serve equally well. Consider changing "sub-region" to "region" throughout.

118     short-wave IR >> short-wave IR (SWIR)

122     Suggest you add "(used in the Normalized Difference Snow Index)" to tie this back to the abstract. So it would read "…SWRI band 6 (used in the Normalized Difference Snow Index)…"

167-170         This isn't clear. Does it mean that six of the scenes that Kern classified were used by the authors to validate their method? Or does it mean that six of the scenes that Kern classified are being offered to readers in the supplement, so that readers can evaluate the author's method? I think it means the former. A re-written sentence might read something like this: "In order to evaluate the classification method suggested by our study [to distinguish it from "this study" used earlier for Kern's study] we processed Landsat 8 reflectance from six clear-sky scenes that Kern (2021) had classified, and then compare results." Then, point readers to where those comparisons can be found (Section 3.2?)

In the caption for Figure S1, it looks like "left of each panel" and "right of each panel" are reversed. Also, where it says "…and the reference classification map (left of each panel) are provided", consider changing to " …and the reference classification map **that our method produced** (**right** of each panel) are **shown**",

264     Classification of a Landsat-8 pixel into ice and open water >> Classification of a Landsat-8 pixel **as** ice **or** open water

276     Remove "in order"

282     Remove "steps"

301     are not consisted solely >> do not consist solely

321     are not fully concentrated by Landsat-8 pixels >> are not entirely filled by Landsat-8 pixels

331     each twelve >> all twelve

366-368     As mentioned in section 3.3, Landsat-8 SIC can be largely deviated from actual SIC if Landsat-8 measures partially-covered grid cell, in other words, SIC computed from partially-covered grid cells may not be representative of actual ice coverage over the entire grid cell  >> As mentioned in section 3.3, SIC computed from partially-covered grid cells may not be representative of actual ice coverage over the entire grid cell

391 consider adding "along with the mean and standard deviation of sea ice concentration" after "…shown in Fig. 6"

415     Should "..estimated over the pixels with such wrongly-masked pixel.." be "estimated **for grid cells** with such wrongly-masked pixel**s** …"  ?

424     are well corresponding >> correspond well

443     sub-range >> range for line 443 and also in the figure caption

445     The contribution of the two threshold variables to $\sigma_{SIC}$ was found that $\rho5$ threshold explains most of … >> Still, the $\rho5$ threshold explains most of …

446     In spite of the relatively high uncertainty in Landsat-8 SIC ranged from 20% to 80%, >> In spite of the relatively high uncertainty in Landsat-8 SIC between 20% and 80%,

460     …category, sub-section with 100% cloud cover based on visual inspection, but less than 100% cloud cover from CFMask was selected. From the collected sub-sections, the $\rho5$ and NDSI values were collected …  >> …category, **those having** 100% cloud cover based on visual inspection, but less than 100% cloud cover from CFMask **were** selected. From **these images**, the $\rho5$ and NDSI values were collected …

472     …and thus SICs produced …   >> …and thus **for** SICs produced …

482      "Chart" should be plural: "Charts"

495     The spreads >>The spread

498     "SIC from the ice chart was found to be positively biased to Landsat-8 SIC,"  Would it be more clear to say ""SIC from ice chart**s** tends to be higher than that found using Landsat-8 SIC, "

528     should be "Although **a** few…"

529     …because melt ponds are not easily discernible to open water, … >> …because melt ponds are not easily **distinguished from** open water, …

533     has robustness >> is robust

617     Comparison of Landsat-8 SIC against SIC retrievals from NASA Team (NT) and Bootstrap (BT) algorithms reveal >> Comparison of Landsat-8 SIC against SIC retrievals from NASA Team (NT) and Bootstrap (BT) algorithms **for two cases** reveal

619     related with >> related to

---

## Referee Comment (RC2)

**Title:**

An Arctic sea ice concentration data record on a 6.25 km polar stereographic grid from three-years' Landsat-8 imagery

**Authors:**

Hee-Sung Jung, Sang-Moo Lee, Joo-Hong Kim, and Kyungsoo Lee

**Main Overview:**

This study focuses on the development on a new high-resolution dataset from Landsat 8 for retrieval of sea ice concentration. There is novelty in this approach compared with more traditional passive microwave methods, and whilst the notion of sea ice concentration retrieval from Optical/IR methods is not intrinscially new, the research presented here demonstrates work of significant value – it is very nice to see research that transitions away from the MODIS sensor (which is approaching end of life and Terra no longer has a regular orbit), and the amount of data processed at high resolution is very novel, and quite exciting! I wholeheartedly agree with the authors thoughtful and well-presented argument of the importance of an independent from PMW SIC dataset given in the introduction.

The code used in this study is also on zenodo, this is really nice to see! I particularly was appreciative of the 'requirements.txt' outlining the specific versions of packages used in this study, this is excellent to see from the perspective of reproducibility. The code is organised with a logical structure and is sufficiently commented and documented for publication.

The dataset is also published as linked on zenodo, which is a suitable archive. The data is in a typical netcdf format with sensible file names, the time parameter is logical with a clearly defined epoch in the metadata, fill values indicated, and it is intuitive and easy to open and plot this data, which looked reasonable when I plotted some data in the Central Arctic.

This is a very good and innovative concept which has been robustly executed. The methodology is sound. On the whole, the manuscript is thorough, but at this stage there are some gaps in justification of particular choices and insufficient detail in the some of the explanations, particularly in regards to the visual inspection component of cloud masking, as I have outlined in my Specific Comments. Therefore, it is my view that this manuscript should be published subject to minor revisions. The manuscript would benefit from some additional proofing, I outline minor typographical corrections where I see them but do not intend for this to be exhaustive.

**Specific Comments:**

General Introduction: There is, of course, already a 1km SIC dataset produced by the University of Bremen based on merging Optical/IR MODIS data with PWM methods. The value of your work is that is truly independent of PWM, and the UoB dataset isn't. Using MODIS in 2024 also has it's own problems. However, I think you need to write a paragraph on this dataset in the intro, and explain the niche of what your dataset does that this one doesn't. Certainly, without reference to this dataset I do not feel your argument can be considered complete.

*Ludwig, V., G. Spreen, & L. T. Pedersen (2020). Evaluation of a New Merged Sea-Ice Concentration Dataset at 1 km Resolution from Thermal Infrared and Passive Microwave Satellite Data in the Arctic. Remote Sens.  12(19), 3183. [doi:10.3390/rs12193183](https://doi.org/10.3390/rs12193183) [Article (PDF file)]*

Line 107: I think there is a better way that 'the rest of this paper' but I like seeing an outline at this point in the paper. I'd probably also do a line break before to enhance clarity.

Line 117: This is probably a good time to mention the 'pole-hole' or unimaged portion of the central Arctic owing to Landsat-8 inclination. You could also do a diagram to illustrate which bit of the Arctic cant be imaged, but I leave this to your judgment. Noted that you explain this in line 130 but it really is quite important for it to go as soon as the sensor is introduced in my opinion.

Line 122: I appreciate why Landsat-8 was chosen over Sentinel 2 given data coverage and robustness of cloud masking. However, I think you need to explicitly explain why you chose Landsat-8, as someone less familiar with Arctic sea ice research (and hasn't had the pleasure of trying to get Sen2Cor working over sea ice!) is going to be scratching their head wondering why you didn't choose Sentinel 2 (given the bands you use are overlapping, Landsat-8 band 5 and 6 approx. equal to S2 8a and 11, and S2 has a better resolution and repeat time). So, my advice here is to add in a couple of sentences explaining why you chose Landsat-8 and make an explicit reference to why you didn't choose Sentinel-2.

Line 127: What method of cloud mask is implemented here? Is it appropriate? Why did you choose 10%?

Section 2.4: I agree with the study choice to use Ice Charts for validation, and I think you have done a good job in conveying why this is the most appropriate choice.

Line 224: Great explanation!

Line 230: The phrase 'a visual inspection' is not exhaustive enough an explanation for this step. I want to know exactly what this process entailed, what images you looked at, what questions you asked, how you looked at them, how many images you looked at, whether anyone else looked at them. I think a diagram of one typical judgement you made would be assistive. I need the thought process here to be outlined to the extent that I could replicate this step and generate the labels myself purely based on what is on the page, and it feels too muddled at the moment for me to plainly understand your meaning.

Line 270: Are these Top of Atmosphere (TOA) or not? In my view it is OK to use TOA for Arctic studies as the aerosol profiles are very poorly constrained and the magnitude of the atmospheric component is small compared to the bright surface, but this should be made very clear. If these are not TOA what processing was applied, and what is the justification for appropriateness?

Line 355: This is robust error propagation and well justified.

Line 380: Thanks for making this clear, it's the right choice and good to see.

Line 461: This is actually quite similar to how MOD29 is derived, which is considered to be one of the better cloud masks over sea ice. I leave it to your judgment if you feel inclusion of that would strengthen or weaken your case, but I think it is robust in any case.

Section 5.2: There does seem to be a notable bias between the ice chart and your dataset at lower SIC%. Can you comment on why this might be?

Line 578-59: always lovely to see DOIs but these should be referenced properly as datasets and cited in text.

Section 5.3: At what point in the season is this Landsat-8 data or is it the whole dataset? Would a more specific analysis around peak melt pond presence perhaps be better? (It is perfectly fine if you do not agree with this but adjust the manuscript to justify what choice was made and why in any case).

**Figures, Tables, and Equations:**

I feel that your captions lack detail throughout. Figures and Tables should have terms used redefined so they can be interpretable independently of the text surrounding it. This will really help improve the clarity of the overall paper and should not be too arduous. You have done a very good job of defining terms within equations – I want to see the same clarity from the captions associated with your Tables and Figures.

Figure 1: Add reference to the mask you used to make this figure. Please also add in the projection and the software you used to make this map. I do not think a scale is necessary on a pan-Arctic map. I would also prefer to see your lat/lon labels appear in-front of the boxes, rather than behind, ie. The 70 deg is partially obscured in the Laptev sea. Please also reiterate the 'study period' with dates, the figure should be as self-contained as possible without reference to the text.

Figure 2: As with figure 1, add a citation for the regional mask and please also add in the projection and the software you used to make this map.

Figure 3: Really like to see diagrams like this, they're very helpful graphical illustrations of your work and improve readership comprehension. Good job!

Table 2: As with figures, tables should be as self-contained as possible. Please add a very brief explanation of C1, C2, C3, and C4 in your caption, as you have with Figure 5.

Figure 4: Redefine $\rho 5$ in your caption please.

Figure 5: Redefine $\rho 5$ in your caption please.

Figure 6: Please add a very brief explanation of C1, C2, C3, and C4 in your caption, as you have with Figure 5.

Figure 7: I think these plots warrant a scale bar, what do you think? I'd also prefer to see the dataset for the true colour images as a reference instead of a hyperlink to a USGS tool.

Figure 8: Define $sigma_{SIC}$ and $\rho 5$ in your caption please.

Table 4: Very quick definition of CFMask please.

Figure 9: Very nice to see error bars and the percentiles these confer to explicit. What R metric is this? Not clear if Pearson/Spearman/Coefficient of Determination or other, so make this explicit in your caption.

Figure 10: I'm not sure at all what this trendline represents, or if it has much physical meaning. Either add context to it in your caption or remove it.

Figure 11: Put the USGS source as a reference. Define specifically what R we are talking about. A scale bar would be nice if it is easy to do, but not essential. Define BT and NT.

Figure 12: What are the units for (b)? Add a definition for 'in' and 'out' in (d).

---

## Author Response (AR1)

**Response to essd-2024-264 RC1**

- The referee's comments are in blue
- The authors' responses are shown in black

The article presents a data set of high resolution and high quality sea ice concentration maps derived from Landsat data that can be used to validate algorithms for ice concentration fields derived from other satellite data. The authors note that the data should find utility as a benchmark data set by which to judge the accuracy of passive microwave-derived ice concentration data. While the spatial extent of each classified Landsat image is quite small relative to a passive microwave ice concentration field, the large number of classified Landsat images should somewhat make up for this limitation.

The data and methods are new, while having been built upon established methods for classifying visible and near-IR data. Methods are described in detail with sufficient references. Error estimates and sources of error are given and discussed in the article. The data are accessible and can be downloaded from the linked Zenodo site. There is one NetCDF file for each of 12 regions. I plotted SIC fields from the Beaufort region file using Panoply, and displayed a number of image days at random. These looked reasonable, but I did not assess the data quality beyond that. I'll note that the summary on the Zenado site needs significant copy editing.

I rate the data set as excellent in terms of its uniqueness, usefulness, and completeness.

The article is well-structured, and overall, the presentation quality is very good, but in some places it lacks clarity. I have made comments under Technical Corrections that may help the authors improve clarity.

The authors sincerely appreciate your time and valuable comments which definitely led to a much better version of the manuscript. In revising the paper, we strove to take up your valuable suggestions and comments and incorporate them into the revision. Please check the authors' responses to the comments below.

**Specific comments**

Section 3.1 says that an addition step in cloud mask quality assessment was to visually compare the cloud mask array from each image with the corresponding true-color image. Please provide more information on this process, including who did the visual inspection and approximately how much time it took for each image. More that 15,000 images are a lot to visually screen, and this step should be described in more detail.

Thank you for pointing this out. The authors agree with your comment and the necessity of more detailed description of the visual inspection process. In the revised manuscript, details of the visual inspection process, including the exact steps taken in the visual inspection process, an example case to show how the visual inspection was executed, who did the inspection, and the time it took to perform the inspection, are added in the appendix. A sentence to point the readers to the appendix was also added in Section 3.1.

[revised manuscript text omitted]

**Technical corrections or wording suggestions**

The authors appreciate the technical corrections and wording suggestions. The authors fully agree with your suggestions in that they can help improve the clarity of the manuscript. Please see the changes made for each of your corrections/suggestions.

**(Line 36) Remove "at least"**

Thank you for your suggestion. In the revised manuscript, "at least" has been removed.

**(Line 54) There have been developed various PMW SIC algorithms >> Various PMW SIC algorithms have been developed**

In the revised manuscript, the suggestion has been incorporated.

**(Line 72) However, there exist discrepancies >> However, discrepancies exist**

In the revised manuscript, 'there exist discrepancies' is changed into 'However, discrepancies exist' by following your suggestion.

**(Line 78) What is meant by "ice/water mixtures"? Note that the differences in algorithms isn't due to the presence of these things, but rather due to the differing sensitivity of the algorithms to these things, so it may be helpful to reword this sentence to reflect that.**

Thank you for your catching of the rather ambiguous phrasing. "Ice/water mixture" was meant to reflect the state of sea ice with melt pond presence which can lead to changes in the emissivity of sea ice. The sentence has been changed to better clarify the meaning of "ice/water mixture" and to reflect that the differences in the retrievals are due to the differing sensitivity of the algorithms.

**[Old]** (Lines 78-79) '…due to the presence of melt ponds and ice/water mixtures, as well as a humid atmosphere…'

**[New]** '…due to the differing sensitivity of retrieval algorithms to the presence of melt pond and the associated emissivity change, as well as a humid atmosphere…'

**(Line 88) It would be helpful add a sentence here that notes that you will use the Kern et al. data to validate your own data as described in Section 2.3.**

Thank you for this helpful suggestion. A sentence mentioning the utility of the dataset by Kern et al. (2022) is added to the revised manuscript after line 88.

**[Added]** The dataset by Kern et al. (2022) is also utilized for validation of the produced Landsat-8 SIC in this study, and the results of the comparison are presented in Section 3.2.

(Line 107) The use of "sub-region" confused me. Here, and for most if not all other occurrences throughout the paper, "region" would serve equally well. Consider changing "sub-region" to "region" throughout.

The authors fully agree with your opinion. In the revised manuscript, all occurrences of "sub-region" were replaced to "region".

(Line 118) short-wave IR >> short-wave IR (SWIR)

Thank you for your catching. "(SWIR)" has been added in the revised manuscript.

(Line 122) Suggest you add "(used in the Normalized Difference Snow Index)" to tie this back to the abstract. So it would read "…SWRI band 6 (used in the Normalized Difference Snow Index)…"

Thank you for your suggestion. This is done in the revised manuscript as the following.

**[Old]** (Line 122) '…SWIR band 6 were used in this study.'

**[New]** '…SWIR band 6 (used in the Normalized Difference Snow Index) were used in this study.'

(Line 167-170) This isn't clear. Does it mean that six of the scenes that Kern classified were used by the authors to validate their method? Or does it mean that six of the scenes that Kern classified are being offered to readers in the supplement, so that readers can evaluate the author's method? I think it means the former. A re-written sentence might read something like this: "In order to evaluate the classification method suggested by our study [to distinguish it from "this study" used earlier for Kern's study] we processed Landsat 8 reflectance from six clear-sky scenes that Kern (2021) had classified, and then compare results." Then, point readers to where those comparisons can be found (Section 3.2?)

Thank you for the comment and suggestion. The sentence (Lines 167-170) in the original manuscript does indeed mean the former as you pointed out. In the revised manuscript, to clarify the unclear meaning of the sentence, this has been changed following your suggestion.

**[Old]** (Lines 167-170) 'In order to evaluate the ice and water classification method (see Section 3.2) suggested by this study, 6 classified scenes (Kern, 2021) under clear sky condition of which scene location and time are provided in the supplements Fig. S1 and Table S1 and the corresponding Landsat-8 reflectivities were used.'

**[New]** 'In order to evaluate the classification method suggested by our study we processed Landsat-8 reflectance from six clear-sky scenes that Kern (2021) had classified, and then compared results. The result of the comparison is presented in Section 3.2 and the location and time of the Landsat-8 scenes that were used in the evaluation are provided in the supplements Fig. S1 and Table S1.'

(Supplements) In the caption for Figure S1, it looks like "left of each panel" and "right of each panel" are reversed. Also, where it says "…and the reference classification map (left of each panel) are provided", consider changing to " …and the reference classification map that our method produced (right of each panel) are shown",

The authors sincerely thank you for this catching. The caption in Figure S1 has been revised following your suggestion as the following.

**[Old]** …True-color images (right of each panel) and the reference classification map (left of each panel) are provided…

**[New]** …True-color images (left of each panel) and the reference classification map that our method produced (right of each panel) are provided…

(Line 264) Classification of a Landsat-8 pixel into ice and open water >> Classification of a Landsat-8 pixel as ice or open water

Thank you for this suggestion. The sentence has been changed following your suggestion in the revised manuscript.

(Line 276) Remove "in order"

Thank you. In the revised manuscript, "in order" has been removed.

(Line 282) Remove "steps"

Thank you for this suggestion. In the revised manuscript, "steps" has been removed.

(Line 301) are not consisted solely >> do not consist solely

Thank you for your suggestion. In the revised manuscript, 'are not consisted solely' is changed into 'do not consist solely'.

(Line 321) are not fully concentrated by Landsat-8 pixels >> are not entirely filled by Landsat-8 pixels

Thank you very much for your suggestion. 'are not fully concentrated by Landsat-8 pixels' is modified by 'are not entirely filled by Landsat-8 pixels' in the revised manuscript.

(Line 331) each twelve >> all twelve

Thank you for the correction. In the revised manuscript, your correction is incorporated.

(Line 366-368) As mentioned in section 3.3, Landsat-8 SIC can be largely deviated from actual SIC if Landsat-8 measures partially-covered grid cell, in other words, SIC computed from partially-covered grid cells may not be representative of actual ice coverage over the entire grid cell >> As mentioned in section 3.3, SIC computed from partially-covered grid cells may not be representative of actual ice coverage over the entire grid cell

Thank you for your suggestion which led to the more concise phrasing. In the revised manuscript, the phrase has been changed by following your suggestion.

(Line 391) consider adding "along with the mean and standard deviation of sea ice concentration" after "…shown in Fig. 6"

Thank you for your suggestion. In the revised manuscript, your suggestion has been reflected as the following:

[Old] (Line 391) '…shown in Fig. 6 (see Table S9 in the supplementary for values).'

[New] '…shown in Fig. 6 along with the mean and standard deviation of SIC (see Table S9 in the supplementary for values).'

(Line 415) Should "..estimated over the pixels with such wrongly-masked pixel.." be "estimated for grid cells with such wrongly-masked pixels …" ?

The authors appreciate your catching. Your description is indeed more accurate than what was in the original manuscript. This has been changed following your suggestion in the revised manuscript.

[Old] (Line 415) '…estimated over the pixels with such wrongly-masked pixel…'

[New] '…estimated for grid cells with such wrongly-masked pixels…'

(Line 424) are well corresponding >> correspond well

Thank you for your suggestion. In the revised manuscript, 'correspond well' can be found rather than 'are well corresponding'.

(Line 443) sub-range >> range for line 443 and also in the figure caption

Thank you for your comment. The suggestion is fully incorporated in the revised manuscript.

(Line 445) The contribution of the two threshold variables to $\sigma$SIC was found that $\rho 5$ threshold explains most of … >> Still, the $\rho 5$ threshold explains most of …

The authors thank you for your valuable suggestion which led to a more concise phrasing of the sentence. In the revised manuscript, your suggestion has been reflected.

(Line 446) In spite of the relatively high uncertainty in Landsat-8 SIC ranged from 20% to 80%, >> In spite of the relatively high uncertainty in Landsat-8 SIC between 20% and 80%,

In the revised manuscript, 'In spite of the relatively high uncertainty in Landsat-8 SIC ranged from 20% to 80%,' has been changed into 'In spite of the relatively high uncertainty in Landsat-8 SIC between 20% and 80%,' by following your suggestion.

(Line 460) …category, sub-section with 100% cloud cover based on visual inspection, but less than 100% cloud cover from CFMask was selected. From the collected sub-sections, the ρ5 and NDSI values were collected … >> …category, those having 100% cloud cover based on visual inspection, but less than 100% cloud cover from CFMask were selected. From these images, the ρ5 and NDSI values were collected …

Thank you for your suggestion. The authors totally agree that the latter sentence better clarifies the process of testing the unmasked cloud pixels. In the revised manuscript, therefore, the following correction has been made:

[Old] (Lines 460-462) '…category, sub-section with 100% cloud cover based on visual inspection, but less than 100% cloud cover from CFMask was selected. From the collected sub-sections, the ρ5 and NDSI values were collected …'

[New] '…category, those having 100% cloud cover based on visual inspection, but less than 100% cloud cover from CFMask were selected. From these images, the ρ5 and NDSI values were collected …'

(Line 472) …and thus SICs produced … >> …and thus for SICs produced …

Thank you for your correction. This has been corrected in the revised manuscript.

(Line 482) "Chart" should be plural: "Charts"

Thank you for your correction. This has been corrected in the revised manuscript.

(Line 495) The spreads >>The spread

Thank you for your correction. This has been corrected in the revised manuscript.

(Line 498) "SIC from the ice chart was found to be positively biased to Landsat-8 SIC," Would it be more clear to say ""SIC from ice charts tends to be higher than that found using Landsat-8 SIC, "

Thank you for your suggestion. The authors agree that the meaning of the sentence is better clarified in the latter. Your suggestion has been incorporated in the revised manuscript.

(Line 528) should be "Although a few…"

Thank you for your correction. This has been corrected in the revised manuscript.

(Line 529) …because melt ponds are not easily discernible to open water, … >> …because melt ponds are not easily distinguished from open water, …

Thank you for your suggestion. This has been changed by following your suggestion in the revised manuscript.

(Line 533) has robustness >> is robust

Thank you for your suggestion. This has been changed by following your suggestion in the revised manuscript.

(Line 617) Comparison of Landsat-8 SIC against SIC retrievals from NASA Team (NT) and Bootstrap (BT) algorithms reveal >> Comparison of Landsat-8 SIC against SIC retrievals from NASA Team (NT) and Bootstrap (BT) algorithms for two cases reveal

Thank you for the detailed correction. What is shown in Section 5.4 is indeed limited to two cases of the PMW retrievals. This has been corrected by following your suggestion in the revised manuscript.

(Line 619) related with >> related to

Thank you for your correction. This has been corrected in the revised manuscript.

**Response to essd-2024-264 RC2**

-     The referee's comments are in blue
-     The authors' responses are shown in black

This study focuses on the development on a new high-resolution dataset from Landsat 8 for retrieval of sea ice concentration. There is novelty in this approach compared with more traditional passive microwave methods, and whilst the notion of sea ice concentration retrieval from Optical/IR methods is not intrinscially new, the research presented here demonstrates work of significant value – it is very nice to see research that transitions away from the MODIS sensor (which is approaching end of life and Terra no longer has a regular orbit), and the amount of data processed at high resolution is very novel, and quite exciting! I wholeheartedly agree with the authors thoughtful and well-presented argument of the importance of an independent from PMW SIC dataset given in the introduction.

The code used in this study is also on zenodo, this is really nice to see! I particularly was appreciative of the 'requirements.txt' outlining the specific versions of packages used in this study, this is excellent to see from the perspective of reproducibility. The code is organised with a logical structure and is sufficiently commented and documented for publication.

The dataset is also published as linked on zenodo, which is a suitable archive. The data is in a typical netcdf format with sensible file names, the time parameter is logical with a clearly defined epoch in the metadata, fill values indicated, and it is intuitive and easy to open and plot this data, which looked reasonable when I plotted some data in the Central Arctic.

This is a very good and innovative concept which has been robustly executed. The methodology is sound. On the whole, the manuscript is thorough, but at this stage there are some gaps in justification of particular choices and insufficient detail in the some of the explanations, particularly in regards to the visual inspection component of cloud masking, as I have outlined in my Specific Comments. Therefore, it is my view that this manuscript should be published subject to minor revisions. The manuscript would benefit from some additional proofing, I outline minor typographical corrections where I see them but do not intend for this to be exhaustive.

The authors sincerely appreciate for your positive evaluation of the manuscript and thank you for your detailed and constructive comments, which definitely helped the manuscript to be more complete in terms of justification of the particular choices made during the process of SIC production/validation and details of the processes undertaken to produce the Landsat-8 SIC dataset. Please check the response to the comments below.

**Specific Comments:**

General Introduction: There is, of course, already a 1km SIC dataset produced by the University of Bremen based on merging Optical/IR MODIS data with PWM methods. The value of your work is that is truly independent of PWM, and the UoB dataset isn't. Using MODIS in 2024 also has it's own problems. However, I think you need to write a paragraph on this dataset in the intro, and explain the niche of what your dataset does that this one doesn't. Certainly, without reference to this dataset I do not feel your argument can be considered complete.

Ludwig, V., G. Spreen, & L. T. Pedersen (2020). Evaluation of a New Merged Sea-Ice Concentration Dataset at 1 km Resolution from Thermal Infrared and Passive Microwave Satellite Data in the Arctic. Remote Sens. 12(19), 3183. doi:10.3390/rs12193183 [Article (PDF f ile)]

The authors sincerely thank you for your suggestion and sharing this great reference with us, which helped to make a strengthened introduction. The authors fully agree that for a complete argument of the necessity of a "fully independent" reference SIC, a discussion of the MODIS/AMSR2 merged SIC product should be included. In the revised manuscript, a paragraph discussing the dataset by Ludwig et al. (2020) and the need for a fully independent dataset despite the presence of an existent high-resolution and pan-Arctic SIC dataset was included.

**[Added]** 'Recently, efforts to leverage the advantages of both VIS/IR sensors and PMW sensors for retrieving SIC have been explored through data merging techniques. Ludwig et al. (2020) used a combination of MODIS and AMSR2 measurements to construct high-resolution (1 km) and spatially continuous SIC data over pan-Arctic areas. This approach exploited the benefits of the 1 km resolution MODIS imagery while mitigating its inherent disadvantage of spatial discontinuity due to clouds by introducing the AMSR2 measurements. While the SIC dataset produced by Ludwig et al. (2020) is both high-resolution and covers pan-Arctic areas, due to the retrievals being dependent on the AMSR2 measurements, the product cannot be considered a fully independent reference data for PMW SIC validation. Therefore, it is still necessary to construct a dataset of Arctic SIC that is fully independent of PMW measurements.'

Reference:

Ludwig, V., Spreen, G., and Pedersen, L. T.: Evaluation of a New Merged Sea-Ice Concentration Dataset at 1km Resolution from Thermal Infrared and Passive Microwave Satellite Data in the Arctic, Remote Sens., 12(19), 3183, doi: https://doi.org/10.3390/rs12193183, 2020.

Line 107: I think there is a better way that 'the rest of this paper' but I like seeing an outline at this point in the paper. I'd probably also do a line break before to enhance clarity.

Thank you for this comment. Following your suggestion, we have rephrased 'the rest of this paper' into 'the remaining sections of this paper are' and have added a line break to enhance clarity in the revised manuscript.

**[Old]** (Line 107) 'The rest of this paper is organized as follows:…'

**[New]** 'The remaining sections of this paper are organized as follows:'

Line 117: This is probably a good time to mention the 'pole-hole' or unimaged portion of the central Arctic owing to Landsat-8 inclination. You could also do a diagram to illustrate which bit of the Arctic cant be imaged, but I leave this to your judgment. Noted that you explain this in line 130 but it really is quite important for it to go as soon as the sensor is introduced in my opinion.

The authors appreciate your valuable suggestions which led to a much-improved visibility and clarity of Fig. 1. In the revised manuscript, illustration of the 'pole-hole' is added as the hatched area in Fig. 1 along with the caption explaining the pole hole. In order to incorporate your comment regarding the timing of bringing up the pole hole, in the revised manuscript, the authors restructured the paragraph to mention the pole hole right after the introduction of the OLI sensor as the following:

**[Removed]** (Line 130) 'It should also be noted that, due to the sun-synchronous orbit of Landsat-8 along with the narrow swath-width of OLI, Landsat-8 does not measure areas where latitude is higher than 82°N.'

**[Added]** '…with a spatial resolution of 30 m. It should be noted that areas with a latitude higher than 82°N in the northern hemisphere are not measured by Landsat-8 (i.e., the hatched area in Fig. 1) due to the orbit inclination of Landsat-8 and the relatively narrow swath width of the OLI.'

[Figure]

Figure 1: Footprints of the collected Landsat-8 images over each region of the pan-Arctic areas during the period of Jan. 2020 – Dec. 2022. The hatched region denotes the areas unmeasured by Landsat-8 due to its orbital inclination (i.e., pole hole). The regions of the pan-Arctic areas were distinguished using the region mask provided by Meier and Stewart (2023). The map projection is NSIDC Sea Ice Polar Stereographic North (EPSG: 3413) and the map was plotted using Python.

Line 122: I appreciate why Landsat-8 was chosen over Sentinel 2 given data coverage and robustness of cloud masking. However, I think you need to explicitly explain why you chose Landsat-8, as someone less familiar with Arctic sea ice research (and hasn't had the pleasure of trying to get Sen2Cor working over sea ice!) is going to be scratching their head wondering why you didn't choose Sentinel 2 (given the bands you use are overlapping, Landsat-8 band 5 and 6 approx. equal to S2 8a and 11, and S2 has a better resolution and repeat time). So, my advice here is to add in a couple of sentences explaining why you chose Landsat-8 and make an explicit reference to why you didn't choose Sentinel-2.

Thank you for mentioning this. The authors agree that an explanation for making the specific choice of Landsat-8 over other available high-resolution measurements, such as Sentinel-2, would help justify this choice. To do this, an explanation of the more robust cloud mask of the Landsat-8 product relative to the Sentinel-2 product was given along with two citations supporting this argument (Zhu et al., 2015; Tarrio et al., 2020).

[Added] 'It is worth noting that the methods developed in this study (described in Section 3) utilize the NIR and SWIR bands for SIC retrieval and are therefore applicable to a wider range of high-resolution sensors that observe at similar bands, including the Multi-Spectral Instrument (MSI) onboard Sentinel-2. However, due to the more robust cloud mask performance of the Landsat-8 product, in this study, the Landsat-8 Collection 2 Level 1 product was selected to be used for the production of reference SIC data (Zhu et al., 2015; Tarrio et al., 2020).'


Line 127: What method of cloud mask is implemented here? Is it appropriate? Why did you choose 10%?

The method of cloud mask implemented here is CFMask, which is the official cloud masking algorithm that generates the quality assessment band (i.e. an array containing masking information for clouds, cloud shadows, cirrus, and fill values) of Landsat-8.

The 10% threshold was given as a search criterion upfront to minimize the acquisition of cloud contaminated scenes. Admittedly, it would be ideal to give a 0% threshold as a criterion upfront for cloud cover, however, this leads to the removal of too many Landsat-8 scenes with clear sky portions available for SIC retrieval (Fig. S1, Table S1). In addition, since cloud pixels are removed from the application of CFMask prior to SIC production, the threshold for the Landsat-8 cloud cover during the acquisition of the data was relaxed to 10%.

Justification of the 10% threshold as well as the method of cloud mask implemented in this study was added in the revised manuscript. Fig. S1 and Table S1, which show the relationship between the number of available Landsat-8 scenes and the cloud cover threshold, were added

in the supplements.

**[Old]** 'only Landsat-8 images with less than 10% daytime cloud cover (solar elevation higher than 15°) were collected.'

**[New]** 'only Landsat-8 images with less than 10% cloud cover (based on fractional cloud masked area from the quality assessment band of Landsat-8) during daytime (solar elevation higher than 15°) were collected. While the threshold of 0% cloud cover would ensure the acquisition of the least cloudy scenes, this also results in the loss of a considerable number of Landsat-8 scenes that contain clear-sky portions (see Fig. S1 and Table S1 in the supplements for the number of available Landsat-8 scenes subject to different threshold values of cloud cover). Therefore, the threshold value for cloud cover was relaxed to 10% during the acquisition of Landsat-8 images.'

**[Added]** (Supplements) Figure S1 and Table S1 Number of available Landsat-8 images subject to varying cloud cover thresholds

[Figure]

**Figure S1: Number of available Landsat-8 images (y-axis) subject to values of different cloud cover thresholds (x-axis) during the period of Jan. 2020 – Dec. 2022 over pan-Arctic areas.**

| Cloud cover threshold | 0 | 5 | 10 | 15 | 20 |
|---|---|---|---|---|---|
| Number of available Landsat-8 scenes | 5,231 | 11,787 | 15,286 | 18,177 | 21,060 |

**Table S1: Number of available Landsat-8 images subject to values of different cloud cover thresholds during the period of Jan. 2020 – Dec. 2022 over pan-Arctic areas.**

Section 2.4: I agree with the study choice to use Ice Charts for validation, and I think you have done a good job in conveying why this is the most appropriate choice.

The authors thank you for your support in our choice of validation data.

Line 224: Great explanation!

Thank you for the positive comment.

Line 230: The phrase 'a visual inspection' is not exhaustive enough an explanation for this step. I want to know exactly what this process entailed, what images you looked at, what questions you asked, how you looked at them, how many images you looked at, whether anyone else looked at them. I think a diagram of one typical judgement you made would be assistive. I need the thought process here to be outlined to the extent that I could replicate this step and generate the labels myself purely based on what is on the page, and it feels too muddled at the moment for me to plainly understand your meaning.

Thank you for pointing this out. The authors agree with your comment and the necessity of a more detailed description of the visual inspection process. In the revised manuscript, details of the visual inspection process, including the exact steps taken in the visual inspection process, an example case to show how the visual inspection was executed, who did the inspection, and the time it took to perform the inspection, are added in the appendix. A sentence to point the readers to the appendix was also added in Section 3.1.

[revised manuscript text omitted]

Line 270: Are these Top of Atmosphere (TOA) or not? In my view it is OK to use TOA for Arctic studies as the aerosol profiles are very poorly constrained and the magnitude of the atmospheric component is small compared to the bright surface, but this should be made very clear. If these are not TOA what processing was applied, and what is the justification for appropriateness?

Thank you for pointing this out. The values of $Q_{DN}$ are values at TOA. This was made explicitly in the revised manuscript.

[Old] '…and $Q_{DN}$ is the reflectivity…'

[New] '…and $Q_{DN}$ is the TOA reflectivity…'

Line 355: This is robust error propagation and well justified.

Thank you for the positive comment.

Line 380: Thanks for making this clear, it's the right choice and good to see.

Thank you for the support of our choice.

Line 461: This is actually quite similar to how MOD29 is derived, which is considered to be one of the better cloud masks over sea ice. I leave it to your judgment if you feel inclusion of that would strengthen or weaken your case, but I think it is robust in any case.

Thank you for mentioning this. However, for this case, the authors feel it would be better to not include a discussion about the MOD29 for the sake bringing more focus to the validation results presented in Section 5.1 rather than the methods used to perform the validation.

Section 5.2: There does seem to be a notable bias between the ice chart and your dataset at lower SIC%. Can you comment on why this might be?

Thank you for this comment. There is indeed a bias in that the ice chart SICs are notably higher than the Landsat-8 SICs. This is one of the known characteristics of ice chart SICs as pointed out in the references by Tonboe et al. (2016) and Cheng et al. (2020), which were used in the original manuscript to explain the bias observed in Fig. 9.

-   Tonboe et al. (2016) reports the overestimation of SIC by ice charts when compared with SIC retrievals from PMW measurements (most pronounced in the lower SIC range) and attributes this bias to "better-safe-than-sorry" approach often taken by the ice charting community.
-   Cheng et al. (2020) also reports the overestimation of ice concentrations by ice analysts when compared to ice concentrations derived from automatic image segmentation.

In order to better explain this bias within the manuscript, comments regarding the higher bias at lower SIC range were inserted in the revised manuscript. In addition, possible causes of this bias, which are discussed in the studies of Tonboe et al. (2016) and Cheng et al. (2020), were also incorporated into the revised manuscript.

**[Old]** 'In addition, SIC from the ice chart was found to be positively biased to Landsat-8 SIC which is also supported by previous works of Tonboe et al. (2016) and Cheng et al. (2020).'

**[New]** 'In addition, SIC from the ice charts tends to be higher than that found from Landsat-8 SIC and the bias is more pronounced in the lower SIC range. This type of state-dependent overestimation of SIC from ice charts has been reported in previous works of Tonboe et al. (2016) and Cheng et al. (2020), which shows that overestimation of SIC from ice charts is largest in the lower SIC range due to the "better-safe-than-sorry" practices of the ice charting community.'

Thank you for this catching. The in-text citations of each dataset are included in the revised manuscript.

Section 5.3: At what point in the season is this Landsat-8 data or is it the whole dataset? Would a more specific analysis around peak melt pond presence perhaps be better? (It is perfectly fine if you do not agree with this but adjust the manuscript to justify what choice was made and why in any case).

Thank you for this comment. The analysis presented in Section 5.3 was performed over six Landsat-8 scenes that were acquired over Jul. 2020, Aug. 2020, and Jul. 2021, which correspond to the melting season in the Arctic. A table showing the Landsat-8 scenes used in the evaluation is provided in Section S4 of the supplements. In the revised manuscript, a sentence pointing to Section 2.5, where the method used to select the six Landsat-8 scenes is explained, is added as well as the time periods of the scenes.

| Filename (Melt Pond Fraction) | Filename (Landsat-8) | Time Difference [sec[ |
|---|---|---|
| 20210719_T45XVK_s2_mpf.nc | LC08_L1TP_173002_20210719_20210729_02_T1 | -1539 |
| 20200806_T31XDL_s2_mpf.nc | LC08_L1GT_023243_20200806_20200916_02_T2 | 7268 |
| 20200806_T31XDL_s2_mpf.nc | LC08_L1TP_023244_20200806_20200916_02_T1 | 7292 |
| 20200711_T13XEL_s2_mpf.nc | LC08_L1TP_065001_20200711_20200912_02_T1 | -5904 |
| 20210719_T14XMQ_s2_mpf.nc | LC08_L1TP_068001_20210719_20210729_02_T1 | -2378 |
| 20200711_T13XEL_s2_mpf.nc | LC08_L1TP_081244_20200711_20200912_02_T1 | 5842 |

**Table S4: List of Landsat-8 scenes and melt pond fraction datasets used for evaluation. Time differences between the two data for each scene are also tabulated.**

**[Added]** 'As mentioned in Section 2.5, a total of six Landsat-8 scenes obtained from the periods of Jul. 2020, Aug. 2020, and Jul. 2021 were used in the evaluation.'

The method used to select the dataset for evaluation is described in detail in Section 2.5 of the original manuscript. The reason for using six datasets was because of the availability of the melt pond fraction dataset when subject to spatiotemporal collocation with the Landsat-8 SIC dataset. In the revised manuscript, this was rephrased to indicate that the data selection was not arbitrary, but rather limited by availability.

**[Old]** 'In this study, the total of six MPF datasets that are spatially overlapped with the coverage of Landsat-8 SIC dataset and have time difference of less than 3 hours with the Landsat-8 scene were selected for evaluation of variation in Landsat-8 SIC due to melt pond presence. The list of selected MPF datasets and the corresponding Landsat-8 scenes can be seen in Table S3 of the supplements.'

**[New]** 'In this study, each MPF dataset was tested for spatiotemporal overlap (time difference of less than 3 hours) with the coverage of Landsat-8 SIC. The total of six MPF datasets were found to be overlapping with the coverage of Landsat-8 SIC and thus available for use in the evaluation. The list of available MPF datasets and the corresponding Landsat-8 scenes can be seen in Table S4 of the supplements.'

**Figures, Tables, and Equations:**

I feel that your captions lack detail throughout. Figures and Tables should have terms used redefined so they can be interpretable independently of the text surrounding it. This will really help improve the clarity of the overall paper and should not be too arduous. You have done a very good job of defining terms within equations – I want to see the same clarity from the captions associated with your Tables and Figures.

The authors appreciate your comments regarding lacking information in the captions and wholeheartedly agree that the caption of the figures need to be edited for better clarity. In the revised manuscript, all the captions have been revised. Please look through the details of the changes made below.

Figure 1: Add reference to the mask you used to make this figure. Please also add in the projection and the software you used to make this map. I do not think a scale is necessary on a pan-Arctic map. I would also prefer to see your lat/lon labels appear in-front of the boxes, rather than behind, ie. The 70 deg is partially obscured in the Laptev sea. Please also reiterate the 'study period' with dates, the figure should be as self-contained as possible without reference to the text.

Thank you for the detailed comments. The revised caption now incorporates the reference of the region masks, the map projection, the software used to make the map, and specification of the 'study period'.

The 70 deg, which was obscured by the boxes, has been moved to the front of the boxes, and a hatched area corresponding to the pole hole was added to the figure.

**[Old]** Caption of Figure 1: 'Footprints of the collected Landsat-8 images over each sub-region over the pan-Arctic areas during the study period.'

**[New]** 'Footprints of the collected Landsat-8 images over each region of the pan-Arctic areas during the period of Jan. 2020 – Dec. 2022. The hatched region denotes the areas unmeasured by Landsat-8 due to its orbital inclination (i.e. pole hole). The regions of the pan-Arctic areas were distinguished using the region mask provided by Meier and Stewart (2023). The map projection is NSIDC Sea Ice Polar Stereographic North (EPSG: 3413) and the map was plotted using Python.'

The old and new version of Fig. 1 can be seen as follows:

[Figure]

| **[Old]** | **[New]** |
|---|---|
| | Figure 1: Footprints of the collected Landsat-8 images over each region of the pan-Arctic areas during the period of Jan. 2020 – Dec. 2022. The hatched region denotes the areas unmeasured by Landsat-8 due to its orbital inclination (i.e., pole hole). The regions of the pan-Arctic areas were distinguished using the region mask provided by Meier and Stewart (2023). The map projection is NSIDC Sea Ice Polar Stereographic North (EPSG: 3413) and the map was plotted using Python. |
| Figure 1: Footprints of the collected Landsat-8 images over each sub-region over the pan-Arctic areas during the study period. | |

Figure 2: As with figure 1, add a citation for the regional mask and please also add in the projection and the software you used to make this map.

As was done with Fig. 1, the reference, the map projection, and the software used to make the map were added to the caption.

**[Old]** Caption of Fig. 2: 'Geographic distribution of the designated sub-regions of the Arctic Ocean, based on NSIDC Sea Ice Region Mask data.'

**[New]** 'Geographic distribution of the designated regions of the Arctic Ocean, based on NSIDC Sea Ice Region Mask data (Meier and Stewart, 2023). The map projection is NSIDC Sea Ice Polar Stereographic North (EPSG: 3413) and the map was plotted using Python.'

Figure 3: Really like to see diagrams like this, they're very helpful graphical illustrations of your work and improve readership comprehension. Good job!

Thank you for your encouragement. It's great to know that the diagram helped with readership comprehension.

Table 2: As with figures, tables should be as self-contained as possible. Please add a very brief explanation of C1, C2, C3, and C4 in your caption, as you have with Figure 5.

Thank you for this catching. Explanations of C1, C2, C3, and C4 have been added in the revised caption. In addition, as was done in Fig. 1 and Fig. 2, the 'study period' was specified in the revised caption.

**[Old]** Caption of Table 2: 'The number of Landsat-8 images for the four cloud mask categories over the twelve sub-regions over the Arctic Ocean during the study period.'

**[New]** 'The number of Landsat-8 images for the four cloud mask categories (i.e. C1: underestimated cloud cover, C2: overestimated cloud cover, C3: correctly estimated cloud cover for cloudy sky, and C4: correctly estimated cloud cover for clear sky) over the twelve regions of the Arctic Ocean during the periods of Jan. 2020 – Dec. 2022.'

Figure 4: Redefine ρ5 in your caption please.

Thank you for this comment. Definitions of $\rho_5$ and NDSI (which was also not defined separately in the caption) are added in the revised caption.

**[Old]** Caption of Fig. 4: '…using $\rho_5$ and NDSI criterion.'

**[New]** '…using $\rho_5$ and NDSI criterion, where $\rho_5$ and NDSI are the TOA reflectivity at band 5 of the OLI sensor and the Normalized Difference Snow Index, respectively.'

Figure 5: Redefine ρ5 in your caption please.

As was done with Fig. 5, the definitions of $\rho_5$ and NDSI were added in the revised caption.

**[Old] Caption of Fig. 5:** '…derived from the selected scenes under perturbed thresholds for NDSI (red) and $\rho_5$ (blue).'

**[New]** '…derived from the selected scenes under perturbed thresholds for NDSI (red) and $\rho_5$ (blue), where $\rho_5$ and NDSI are the TOA reflectivity at band 5 of the OLI sensor and the Normalized Difference Snow Index, respectively.'

Figure 6: Please add a very brief explanation of C1, C2, C3, and C4 in your caption, as you have with Figure 5.

Thank you for this comment. Explanations of C1, C2, C3, and C4 were added in the revised caption.

**[Old] Caption of Fig. 6:** '…The black, red, blue, and green bars indicate values for categories C1, C2, C3, and C4, respectively.'

**[New]** '…The black, red, blue, and green bars indicate values for categories C1 (i.e., underestimated cloud cover), C2 (i.e., overestimated cloud cover), C3 (i.e., correctly estimated cloud cover for cloudy sky), and C4 (i.e., correctly estimated cloud cover for clear sky), respectively.'

Figure 7: I think these plots warrant a scale bar, what do you think? I'd also prefer to see the dataset for the true colour images as a reference instead of a hyperlink to a USGS tool.

Thank you for this thoughtful comment. The authors do agree that addition of a scale bar would help with the interpretation of the figure and have added them in the revised figure. In addition, the hyperlink to a USGS tool was replaced with the reference to the dataset in the revised caption.

**[Old]** Caption of Figure 7: '… The true-color images were downloaded from United States Geological Survey Earth Explorer (https://earthexplorer.usgs.gov/, last access: May 22, 2024).'

**[New]** '… The true-color images were obtained from Earth Resources Observation and Science (EROS) Center (2020).'

The revised figures now include the scale length information as:

[Figure]

Figure 7: Example of (a, d, g, i) original Landsat-8 true-color image, (b, e, h, k) classification map of ice (white), open water (blue), and cloud (cyan, purple, and red), and (c, f, i, l) Landsat-8 SICs with 6.25 km resolution on (first row) Mar. 22, 2022 over the Kara Sea, (second row) Mar. 17, 2021 over the Barents Sea, (third row) Jun. 26, 2022 over the Kara Sea, and (fourth row) Jun. 15, 2022 over the Beaufort Sea. From top to bottom row, the select cases correspond to the cloud contamination categories of 1, 2, 3, and 4 respectively. SICs are not estimated over areas of cloud mask (cyan, purple and red pixels in the middle column), and SICs near the coastal area (6.25 km) are masked in this figure. The true-color images were obtained from Earth Resources Observation and Science (EROS) Center (2020).

Figure 8: Define sigmaSIC and ρ5 in your caption please.

Thank you for this comment. In the revised caption, definition of $\sigma_{SIC}$, $\rho_5$, and NDSI were added.

[Old] Caption of Fig. 8: '(a) Uncertainties in Landsat-8 SICs and (b) contributions of the $\rho_5$ (blue) and the NDSI thresholds (red) to the estimated uncertainties for different SIC sub-range....'

[New] '(a) Uncertainties in Landsat-8 SICs ($\sigma_{SIC}$) and (b) contributions of the $\rho_5$ (blue) and the NDSI thresholds (red) to the estimated uncertainties for different SIC sub-range, where $\rho_5$ and NDSI are the TOA reflectivity at band 5 of the OLI sensor and the Normalized Difference Snow Index, respectively....'

Table 4: Very quick definition of CFMask please.

Thank you for this comment. Definition of CFMask is added in the revised caption.

[Old] Caption of Table 4: 'The number of cloud pixels that were undetected from CFMask...'

[New] 'The number of cloud pixels that were undetected from the C Function of Mask (CFMask)...'

Figure 9: Very nice to see error bars and the percentiles these confer to explicit. What R metric is this? Not clear if Pearson/Spearman/Coefficient of Determination or other, so make this explicit in your caption.

Thank you for the positive feedback on the error bars/percentiles. It is nice to hear that they were helpful for the interpretation of the figure. The R metric used in this figure is the Pearson correlation coefficient and in the revised caption, an explanation about the numerical values (including the Pearson correlation coefficient) that appear on the top of Fig. 9a is added.

[Old] Caption of Fig. 9: '...shown as the red vertical lines. (b) For the same SIC intervals...'

[New] '...shown as the red vertical lines. The values for number of data points (N), root-mean-square error (RMSE), bias, and Pearson correlation coefficient (R) are presented. (b) For the same SIC intervals...'

Figure 10: I'm not sure at all what this trendline represents, or if it has much physical meaning. Either add context to it in your caption or remove it.

Thank you for this suggestion. The authors agree that the trendline shown doesn't exhibit much physical meaning in the context of testing robustness of the Landsat-8 retrievals against melt ponds. In the revised figure, the trendline is removed following your suggestion.

The caption about the trend line is also removed and an explanation of the numerical values that are shown on the top-left of the figure is added in the revised caption.

The old and revised figures are found as follows:

| [Old] | [New] |
|---|---|
|
[Figure]
 | |
| Figure 10: Scatter plot of net ice surface fraction (x-axis) and Landsat-8 SIC (y-axis). The data points shown satisfy SIC$_{MPF}$=100% and have MPF that vary from 0% to 33%. Data points with more than 4% deviation of Landsat-8 SIC from 100% ice concentration are highlighted in red. The red line indicates the least-squares regression line. | Figure 10: Scatter plot of net ice surface fraction (x-axis) and Landsat-8 SIC (y-axis). The data points shown satisfy SIC$_{MPF}$=100% and have MPF that vary from 0% to 33%. Data points with more than 4% deviation of Landsat-8 SIC from 100% ice concentration are highlighted in red. The values for number of datapoints (N), Pearson correlation coefficient (R), and p-value for the correlation coefficient are presented. |

And the corresponding captions has been changed as:

**[Old]** Caption of Fig. 10: '…are highlighted in red. The red line indicates the least-squares regression line.'

**[New]** '…are highlighted in red. The values for number of data points (N), Pearson correlation coefficient (R), and p-value for the correlation coefficient are presented.'

Figure 11: Put the USGS source as a reference. Define specifically what R we are talking about. A scale bar would be nice if it is easy to do, but not essential. Define BT and NT.

Thank you for this comment. In the same context as Fig. 7, the authors agree that a scale bar would indeed be helpful for the interpretation of the figure. In the revised figure, scale bars have been added.

As for the captions, similar to what was done for Fig. 7, the USGS source was added as a reference along with the reference for the passive microwave SIC datasets. Definitions of BT, NT, and the numerical values are also added in the revised caption.

**[Old]** Caption of Fig. 11: 'Geographical distributions of (a, i) original Landsat-8 true-color image, (e, m) Landsat-8 SIC, (b, j) SIC from BT algorithm, (f, n) SIC from NT algorithm, (c, k) difference in SICs between BT and Landsat-8, (g, o) difference in SICs between NT and Landsat-8 and scatterplot (d, l) between Landsat-8 SIC and SIC from BT and (h, p) between Landsat-8 SIC and SIC from NT. Upper two panels for July 21, 2022 (melting season) over the Laptev Sea and for March 4, 2020 over the Chukchi Sea, respectively. The true-color images

were downloaded from United States Geological Survey Earth Explorer (https://earthexplorer.usgs.gov/, last access: May 22, 2024).'

[New] 'Geographical distributions of (a, i) original Landsat-8 true-color image, (e, m) Landsat-8 SIC, (b, j) SIC from the Bootstrap (BT) algorithm, (f, n) SIC from the NASA Team (NT) algorithm, (c, k) difference in SICs between BT and Landsat-8, (g, o) difference in SICs between NT and Landsat-8 and scatterplot (d, l) between Landsat-8 SIC and SIC from BT and (h, p) between Landsat-8 SIC and SIC from NT. The values of root-mean-square error (RMSE), bias, and Pearson correlation coefficient (R) are presented with the scatter plots. Upper two panels for July 21, 2022 (melting season) over the Laptev Sea and for March 4, 2020 over the Chukchi Sea, respectively. The true-color images were obtained from Earth Resources Observation and Science (EROS) Center (2020) and the SIC retrievals from the BT and NT algorithms were obtained from Meier et al. (2021).'

The revised figure can be seen as:

[Figure]

Figure 11: Geographical distributions of (a, i) original Landsat-8 true-color image, (e, m) Landsat-8 SIC, (b, j) SIC from the Bootstrap (BT) algorithm, (f, n) SIC from the NASA Team (NT) algorithm, (c, k) difference in SICs between BT and Landsat-8, (g, o) difference in SICs between NT and Landsat-8 and scatterplot (d, l) between Landsat-8 SIC and SIC from BT and (h, p) between Landsat-8 SIC and SIC from NT. The values of root-mean-square error (RMSE), bias, and Pearson correlation coefficient (R) are presented with the scatter plots. Upper two panels for July 21, 2022 (melting season) over the Laptev Sea and for March 4, 2020 over the Chukchi Sea, respectively. The true-color images were obtained from Earth Resources Observation and Science (EROS) Center (2020) and the SIC retrievals from the BT and NT algorithms were obtained from Meier et al. (2021).

Figure 12: What are the units for (b)? Add a definition for 'in' and 'out' in (d).

Thank you for this catching. Since (b) is a map showing the number of Landsat-8 pixels located within each grid cell, the units for (b) should be "Number of Landsat-8 Pixels". This unit was added to the colorbar in the revised figure.

In addition, the "In" and "Out" designating grid cells located inside and outside the sub-region, respectively, were defined in the revised caption.

**[Old]** Caption of Fig. 12: 'Variables in the Landsat-8 SIC netCDF. The scene is from Jun. 12, 2021 over the Canadian Archipelago.'

**[New]** 'Variables in the Landsat-8 SIC netCDF. The scene is from Jun. 12, 2021 over the Canadian Archipelago. For the (d) sub-region mask, 'In' and 'Out' denote grid cells located inside and outside the designated region, respectively.'

The revised figure can be seen as:

[Figure]

Figure 12: Variables in the Landsat-8 SIC netCDF. The scene is from Jun. 12, 2021 over the Canadian Archipelago. For the (d) sub-region mask, 'In' and 'Out' denote grid cells located inside and outside the designated region, respectively.

---

## Author Response (AR2)

**Response to Editor's Comments**

- The referee's comments are in blue
- The authors' responses are shown in black

Thanks to both reviewers for their comprehensive reviews, and to the author team for their detailed responses. I have reviewed these and agree that they are sufficient for publication.

The authors of this study sincerely thank you for all the constructive comments that have led to the completion of a much-improved manuscript. Please check the response to your comments below.

I do have one technical query after reading the revised manuscript:
Appendix B:
"Step 1. Generating jpeg file of cloud mask (i.e., cloud mask image)." - surely the use of a lossy compression format could be detrimental to the cloud mask fidelity, right? Or is "jpeg" incorrect here?

Apologies for the confusion. The cloud mask images used in the visual inspection were made in Portable Network Graphic (PNG) format with lossless data compression.

**[Old]** 'Step 1. Generating jpeg file of cloud mask (i.e., cloud mask image).'

**[New]** 'Step 1. Generating Portable Network Graphics (PNG) file of cloud mask (i.e., cloud mask image).'

Additional private note (visible to authors and reviewers only):
Thank you author team, this response is great. I would like to clarify the above comment though.

Thank you for the positive comment. Your comment has been addressed above.